# pUL21 is a viral phosphatase adaptor that promotes herpes simplex virus replication and spread

Tomasz H. Benedyk[1], Julia Muenzner[1¤a], Viv Connor[1], Yue Han[1], Katherine Brown[1], Kaveesha J. Wijesinghe[1¤b], Yunhui Zhuang[1], Susanna Colaco[1], Guido A. Stoll[1¤c], Owen S. Tutt[1], Stanislava Svobodova[1], Dmitri I. Svergun[2], Neil A. Bryant[3], Janet E. Deane[4], Andrew E. Firth[1], Cy M. Jeffries[2], Colin M. Crump[1]*, Stephen C. Graham[1]*

1 Department of Pathology, University of Cambridge, Cambridge, United Kingdom, 2 European Molecular Biology Laboratory (EMBL) Hamburg Site, Hamburg, Germany, 3 Department of Veterinary Medicine, University of Cambridge, Cambridge, United Kingdom, 4 Cambridge Institute for Medical Research, University of Cambridge, Cambridge Biomedical Campus, Cambridge, United Kingdom

¤a Current address: Charité–Universitätsmedizin Berlin, Institute of Biochemistry, Berlin, Germany
¤b Current address: Department of Chemistry, Faculty of Science, University of Colombo, Sri Lanka
¤c Current address: Molecular Immunity Unit, Department of Medicine, University of Cambridge, MRC Laboratory of Molecular Biology, Cambridge, United Kingdom
* cmc56@cam.ac.uk (CMC); scg34@cam.ac.uk (SCG)

**Data Availability Statement:** The authors confirm that all data underlying the findings are fully available without restriction. The SAXS data measured for each individual concentration, with

## Abstract

The herpes simplex virus (HSV)-1 protein pUL21 is essential for efficient virus replication and dissemination. While pUL21 has been shown to promote multiple steps of virus assembly and spread, the molecular basis of its function remained unclear. Here we identify that pUL21 is a virus-encoded adaptor of protein phosphatase 1 (PP1). pUL21 directs the dephosphorylation of cellular and virus proteins, including components of the viral nuclear egress complex, and we define a conserved non-canonical linear motif in pUL21 that is essential for PP1 recruitment. *In vitro* evolution experiments reveal that pUL21 antagonises the activity of the virus-encoded kinase pUS3, with growth and spread of pUL21 PP1-binding mutant viruses being restored in adapted strains where pUS3 activity is disrupted. This study shows that virus-directed phosphatase activity is essential for efficient herpesvirus assembly and spread, highlighting the fine balance between kinase and phosphatase activity required for optimal virus replication.

## Author summary

Herpes simplex virus (HSV)-1 is a highly prevalent human virus that causes life-long infections. While the most common symptom of HSV-1 infection is orofacial lesions ('cold sores'), HSV-1 infection can also cause fatal encephalitis and it is a leading cause of infectious blindness. The HSV-1 genome encodes many proteins that dramatically remodel the environment of infected cells to promote virus replication and spread, including enzymes that add phosphate groups (kinases) to cellular and viral proteins in order to

an accompanying report are made available in the Small Angle Scattering Biological Data Bank entry SASDKW8. The mass spectrometry proteomic data has been deposited with the ProteomeXchange Consortium (http://www.proteomexchange.org/) via the PRIDE partner repository under the data set identifier PXD027257. Sequencing data have been deposited in the ArrayExpress database (http://www.ebi.ac.uk/arrayexpress) under the accession numbers E-MTAB-10788.

**Funding:** JED is supported by a Senior Research Fellowship from the Wellcome Trust (219447/Z/19/Z). This work was supported by a Senior Research Fellowship from the Wellcome Trust to AEF (106207/Z/14/Z), a Biotechnology and Biological Sciences Research Council (BBSRC) Research Grant to CMC (BB/M021424/1) and a Sir Henry Dale Fellowship, jointly funded by the Wellcome Trust and the Royal Society, to SCG (098406/Z/12/B). The funders had no role in study design, data collection and analysis, decision to publish, or preparation of the manuscript.

**Competing interests:** The authors have declared that no competing interests exist.

fine-tune their function. Here we identify that pUL21 is an HSV-1 protein that binds directly to protein phosphatase 1 (PP1), a highly abundant cellular enzyme that removes phosphate groups from proteins. We demonstrate that pUL21 stimulates the specific dephosphorylation of both cellular and viral proteins, including a component of the viral nuclear egress complex that is essential for efficient assembly of new HSV-1 particles. Furthermore, our *in vitro* evolution experiments demonstrate that pUL21 antagonises the activity of the HSV-1 kinase pUS3. Our work highlights the precise control that herpesviruses exert upon the protein environment within infected cells, and specifically the careful balance of kinase and phosphatase activity that HSV-1 requires for optimal replication and spread.

## Introduction

Herpes simplex virus (HSV)-1 is a highly-prevalent human pathogen that establishes life-long latent infection, with reactivation of virus from peripheral neurons manifesting as orofacial or genital lesions or (occasionally) life-threatening viral encephalitis [1–3]. Like all herpesviruses, HSV-1 dramatically remodels the intracellular environment of infected cells to promote the production and dissemination of progeny virus particles [4,5]. After receiving their genomic cargo in the nucleus, HSV-1 capsids traverse the nuclear envelope via sequential envelopment and de-envelopment steps catalysed primarily by the viral nuclear egress complex (NEC), comprising pUL31 and pUL34 [6]. Cytoplasmic capsids associate with glycoprotein-studded membranes via a proteinaceous layer called *tegument*, budding into the lumen of post-Golgi intracellular vesicles ("secondary envelopment") before being trafficked to and released at the plasma membrane [7,8]. Proteins of the HSV-1 tegument layer play roles in dampening the innate immune response to infection [9,10] and modulating host-cell morphology [11] in addition to their structural roles in virion assembly.

The tegument protein pUL21 exemplifies the multifunctional nature of tegument proteins, being required for efficient capsid nuclear egress, secondary envelopment and virus spread. pUL21 is conserved across alphaherpesvirus [12] and is known to bind the conserved protein pUL16 [13,14], both proteins being known to promote transport of newly-assembled genome-containing virus capsids from the nucleus to the cytoplasm for subsequent envelopment [15–17]. However, the requirement for pUL16 and pUL21 to promote nuclear egress varies depending on the virus strain and cell line studied [16] and they function independently in this process [18]. Specifically, it has been shown that pUL21 is required for the correct 'smooth' distribution of the NEC around the nuclear envelope as viruses lacking pUL21 exhibiting large NEC punctae on the nuclear rim [18]. However, the molecular mechanisms by which pUL21 promotes viral nuclear egress remains uncharacterised. In addition to promoting capsid translocation to the cytoplasm, it has been reported that the pUL21:pUL16 complex may promote wrapping of cytoplasmic virions via interactions with the viral proteins pUL11 and gE [19], although the evidence for this is equivocal [8]. pUL21 has a poorly-characterised role in promoting virus spread, which may involve its interaction with gE [19]. Deletion of pUL21 leads to rapid outgrowth of HSV-1 harbouring mutations in gK that promote syncytia formation, where infected cells fuse with neighbouring cells, and pUL21 is required for maintenance of the syncytial phenotype in strains of HSV-1 where gB is mutated to induce syncytium formation [20]. pUL21 has also been observed to promote the formation of long cellular processes, potentially by regulating microtubule polymerisation [21]. Pseudorabies virus (PrV) pUL21 has been extensively studied as mutations in the gene encoding this protein contribute to the

attenuation of the vaccine strain *Bartha* [22]. PrV pUL21 is required for neurovirulence and a recent study identified that pUL21 binds the dynein light chain protein Roadblock-1, promoting retrograde axonal transport [23].

The diverse range of pUL21 functions and putative binding partners has complicated mechanistic dissection of its function, since abolishing pUL21 expression during infection impairs all functions of this multifaceted protein. Mechanistic insights into pUL21 are further hampered by its lack of homology to any well-characterised protein. HSV-1 pUL21 is a 58 kDa cytoplasmic protein [21] and crystallographic studies showed that pUL21 comprises two well-folded domains separated by a protease-sensitive linker region that is likely to be disordered [24,25]. Neither domain bears significant resemblance to other structurally characterised proteins, preventing inference of function by analogy. However, it was observed that the C-terminal domain of pUL21 co-purifies with RNA following recombinant expression in bacteria, suggesting that nucleic acid binding may represent yet another function of the enigmatic pUL21 protein [25].

We used quantitative proteomics to identify the cellular interaction partners of pUL21, revealing that pUL21 binds the catalytic subunit of protein phosphatase 1 (PP1) and the ceramide transport protein CERT, a protein that is regulated by phosphorylation. We show that pUL21 binds directly to PP1 and CERT, accelerating CERT dephosphorylation both *in vitro* and in cells, and identify a conserved motif in pUL21 required for PP1 binding. HSV-1 mutants where the ability of pUL21 to bind PP1 is abolished have impaired growth and replication, but these mutants rapidly adapt to cell culture via compensatory mutations in the viral kinase gene US3. We show that pUL21 and pUS3 regulate phosphorylation of multiple proteins, including components of the viral NEC, via opposing activities. While viral and cellular kinases have been intensely studied for many decades, the specificity and regulation of phosphatases is much less well defined. Our study demonstrates that a fine balance between kinase and phosphatase activity is required for herpesvirus replication and spread.

## Results

To identify putative cellular binding partners, C-terminally GFP-tagged pUL21 (pUL21-GFP) was immunoprecipitated following ectopic expression in stable isotope labelling of amino acids in cell culture (SILAC)-labelled human embryonic kidney (HEK293T) cells. Quantitative mass spectrometry analysis identified that the catalytic subunit of protein phosphatase 1 (PP1) and the ceramide-transport protein CERT (a.k.a. Collagen type IV alpha-3-binding protein, COL4A3BP, or Goodpasture antigen binding protein, GPBP) were strongly enriched in the pUL21-GFP immunoprecipitation compared with a GFP control (Fig 1A and S1 Data). There are three isoforms of the PP1 catalytic subunit (α, β and γ) in human cells that share very high sequence identity (>85% amino acid identity overall) [26], complicating the definitive identification of co-precipitated isoforms by mass spectrometry, but immunoblots confirmed that all three isoforms co-immunoprecipitate with pUL21-GFP (Fig 1B). GST pull-down experiments using reagents purified following recombinant expression in *Escherichia coli* (pUL21 and PP1γ) and HEK293F cells (CERT) confirmed that pUL21 binds directly to both PP1 and CERT (Fig 1C). Published quantitative viromics analysis [27] confirms that the abundance of CERT and all three PP1 catalytic subunit isoforms is unchanged in keratinocytes over the course of HSV-1 infection.

PP1 is an abundant and highly active cellular phosphatase [26]. The catalytic subunit of PP1 has low intrinsic specificity and substrates are recruited for dephosphorylation via interactions with adaptor proteins known as regulatory subunits (also known as regulatory interactors of protein phosphatase one, RIPPOs) [26,28]. CERT mediates the non-vesicular transport of the

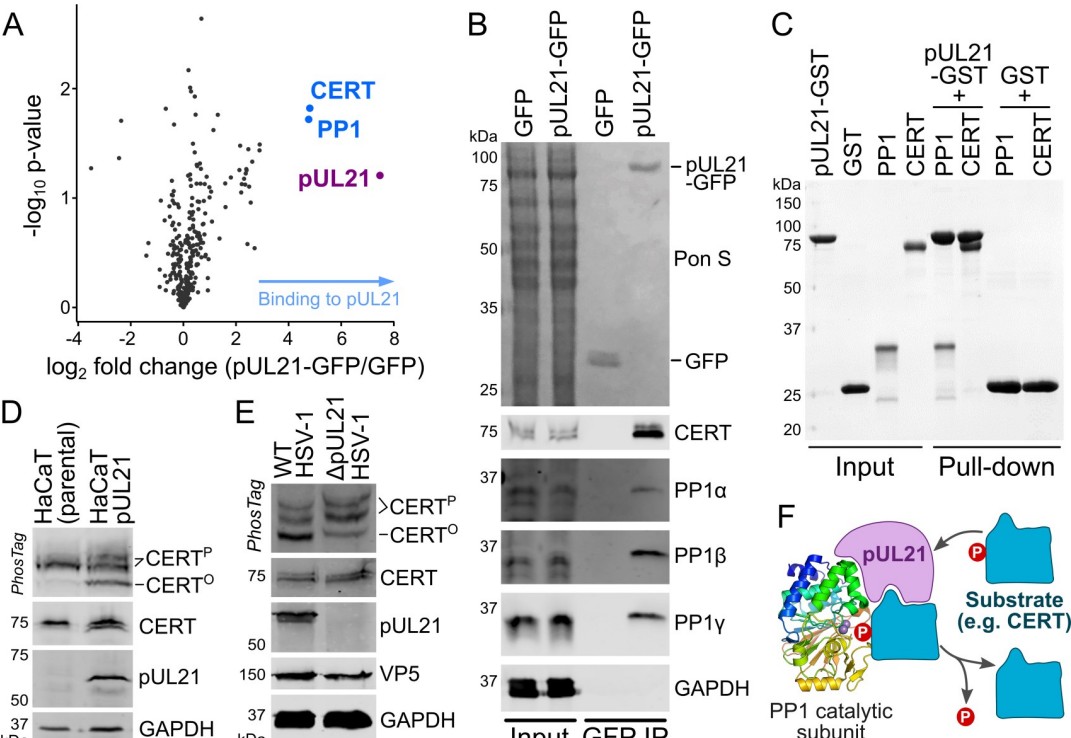

**Fig 1. HSV-1 pUL21 directly binds host proteins CERT and PP1, promoting CERT dephosphorylation.** (**A**) SILAC-labelled HEK293T cells were transfected with plasmids expressing GFP-tagged pUL21 or GFP alone. At 24 hours post-transfection the cells were lysed, subjected to immunoprecipitation (IP) using a GFP affinity resin, captured proteins were proteolytically digested and co-precipitated proteins identified using quantitative mass spectrometry. The horizontal axis shows average fold enrichment in IP of pUL21-GFP compared to GFP across three biological replicates and the vertical axis shows significance (two-sided t-test) across the three replicates. Proteins CERT and PP1 were identified as putative binding partners (blue). (**B**) Unlabelled HEK293T cells were transfected with plasmids expressing GFP-tagged pUL21 or GFP alone and subjected to IP as in (**A**). Captured proteins were subjected to SDS-PAGE and immunoblotting using the antibodies shown. GAPDH is used as a loading control and Ponceau S (Pon S) staining of the immunoblot membrane before blocking shows all proteins transferred. (**C**) Purified recombinant pUL21-GST or GST alone were immobilised on GSH resin and incubated with prey proteins PP1(7–300)-$H_6$ or Strep-CERT. After washing, the bound complexes were eluted and visualized by SDS-PAGE (Coomassie). (**D**) Lysates of HaCaT cells (parental or stably expressing pUL21) were analysed by SDS-PAGE and immunoblotting using the antibodies listed. The upper strip depicts SDS-PAGE where PhosTag reagent was added, retarding the migration of phosphorylated proteins to enhance separation of CERT that is hyper- (CERT$^P$) or hypo-phosphorylated (CERT$^O$). (**E**) HaCaT cells were infected at MOI = 5 with wild-type HSV-1 or a mutant lacking pUL21 expression (ΔpUL21). Lysates were harvested at 16 hours post-infection (hpi) and subjected to SDS-PAGE plus immunoblotting as in (**D**). The HSV-1 major capsid protein VP5 is used as a marker of infection. (**F**) Schematic representation of putative pUL21 activity, recruiting specific substrates for dephosphorylation by the PP1 catalytic subunit.

lipid ceramide from the endoplasmic reticulum (ER) to the *trans*-Golgi, a crucial step in the synthesis of sphingomyelin and derived sphingolipids [29]. Interestingly, the activity of CERT is known to be regulated via phosphorylation: hyperphosphorylated CERT (CERT$^P$) adopts an inactive conformation that is not membrane associated whereas hypophosphorylated CERT (CERT$^O$) associates with ER and *trans*-Golgi membranes to promote ceramide exchange [30–32]. The ability of pUL21 to bind directly to PP1 and CERT suggested that pUL21 may be a novel viral PP1 regulatory subunit, recruiting PP1 to CERT to promote its dephosphorylation and hence activation. To test this hypothesis, the relative abundance of CERT$^P$ and CERT$^O$ was measured in human keratinocyte cells (HaCaT) where pUL21 was stably expressed and in HaCaT cells infected with either wild-type HSV-1 or a mutant where expression of pUL21 had been disrupted (ΔpUL21). In both cases, we observed that the CERT$^O$ predominates when

pUL21 is present (Fig 1D and 1E), consistent with the identification of pUL21 as a viral regulatory subunit that promotes CERT dephosphorylation via direct recruitment of PP1 (Fig 1F).

Most cellular PP1 interacting proteins contain a conserved "RVxF" docking motif (consensus sequence [KR][KR][VI]ψ[FW], where ψ is any amino acid except FIMYDP) plus one or more ancillary motifs [33]. pUL21 does not contain any peptide sequences that conform to this consensus. It does contain two stretches of amino acids, [12]RDVVF[16] and [285]RELWW[289], that conform to a less stringent definition of the RVxF motif ([RK]$x_{0-1}$[VI]{P}[FW], where $x_{0-1}$ is any residue, or no residue at all, and {P} is any residue except proline) [34]. Both of these sequences are in structured regions of the N- and C-terminal domains, respectively, and the key hydrophobic residues are buried in the core of the protein (S1 Fig). As such, these residues could not bind PP1 without significant structural rearrangement. The question thus arises: how does pUL21 associate with PP1?

Before dissecting the molecular determinants of pUL21 binding to PP1 and CERT we sought to determine the conformation of purified pUL21 in solution. Previous studies had shown that the two structured domains of HSV-1 pUL21 are joined by a linker region of 64 amino acids (Fig 2A) that is presumed to be disordered due to its susceptibility to proteolytic cleavage during recombinant purification [25]. We purified full-length C-terminally hexahistidine-tagged pUL21 (pUL21-$H_6$) following recombinant expression in *E. coli* (Fig 2B). While the C-terminal domain of pUL21 had previously been shown to co-purify with bacterial RNA [25] we did not observe nucleic acid co-purification with the full-length protein, pUL21-$H_6$ having a 260:280 nm absorbance ratio of ~0.58. Size-exclusion chromatography with inline multi-angle light scattering (SEC-MALS) confirmed that the protein was predominantly monodisperse and monomeric (Fig 2C). Small-angle X-ray scattering (SAXS) analysis of purified pUL21-$H_6$ (Fig 2D and S1 Table) showed that the protein does not adopt a single compact globular conformation: the frequency of real-space distances within the protein ($p(r)$ profile, Fig 2E) is highly asymmetric with an extended tail of longer distance frequencies spanning the range 10–18 nm. The slow and steady decay of the $p(r)$ profile, with no distinct peak evident at longer distance frequencies, indicates that pUL21-$H_6$ lacks a fixed distance between the N- and C-terminal domains. Moreover, the dimensionless Kratky plot (Fig 2F) lacks the distinct bell-shaped peak at $sR_g = \sqrt{3}$ that is typical of globular proteins [35], indicating that pUL21-$H_6$ is a flexible and/or elongated particle. The ensemble optimisation method (EOM) [36] was therefore employed to characterise the conformational heterogeneity of pUL21. The fitted pUL21-$H_6$ ensemble (Fig 2G, 2H and 2I) samples a wide distribution of states that span compact ($R_g$ = 2.5–3.7 nm, 34% volume fraction), intermediate ($R_g$ = 3.7–5.0 nm, 50% volume fraction) and elongated ($R_g$ > 5 nm, 17% volume fraction) conformations. This is consistent with the pUL21 linker region being conformationally heterogeneous, with N- and C-terminal domains moving toward and away from each other in solution like two ends of an accordion (Fig 2J).

Having validated that the N- and C-terminal domains of pUL21 are joined by a mobile linker region, and can thus be considered as independent functional units, a series of GFP-tagged truncations mutants were designed to identify pUL21 regions that bind CERT and PP1 (Fig 2K). Immunoprecipitation experiments identified that PP1 binding requires both the N-terminal domain and flexible linker (Fig 2L). However, PP1 binding is lost when this region is N-terminally GFP tagged, suggesting that obstruction of the amino terminus of pUL21 via fusion with a bulky GFP domain (approximately the same size as the pUL21 N-terminal domain) could prevent PP1 binding via steric hindrance. Similarly, the C-terminal domain of pUL21 mediates CERT binding but only when the GFP tag is separated from this domain via the flexible linker or is attached to the carboxy terminus of the protein (Fig 2L), suggesting that

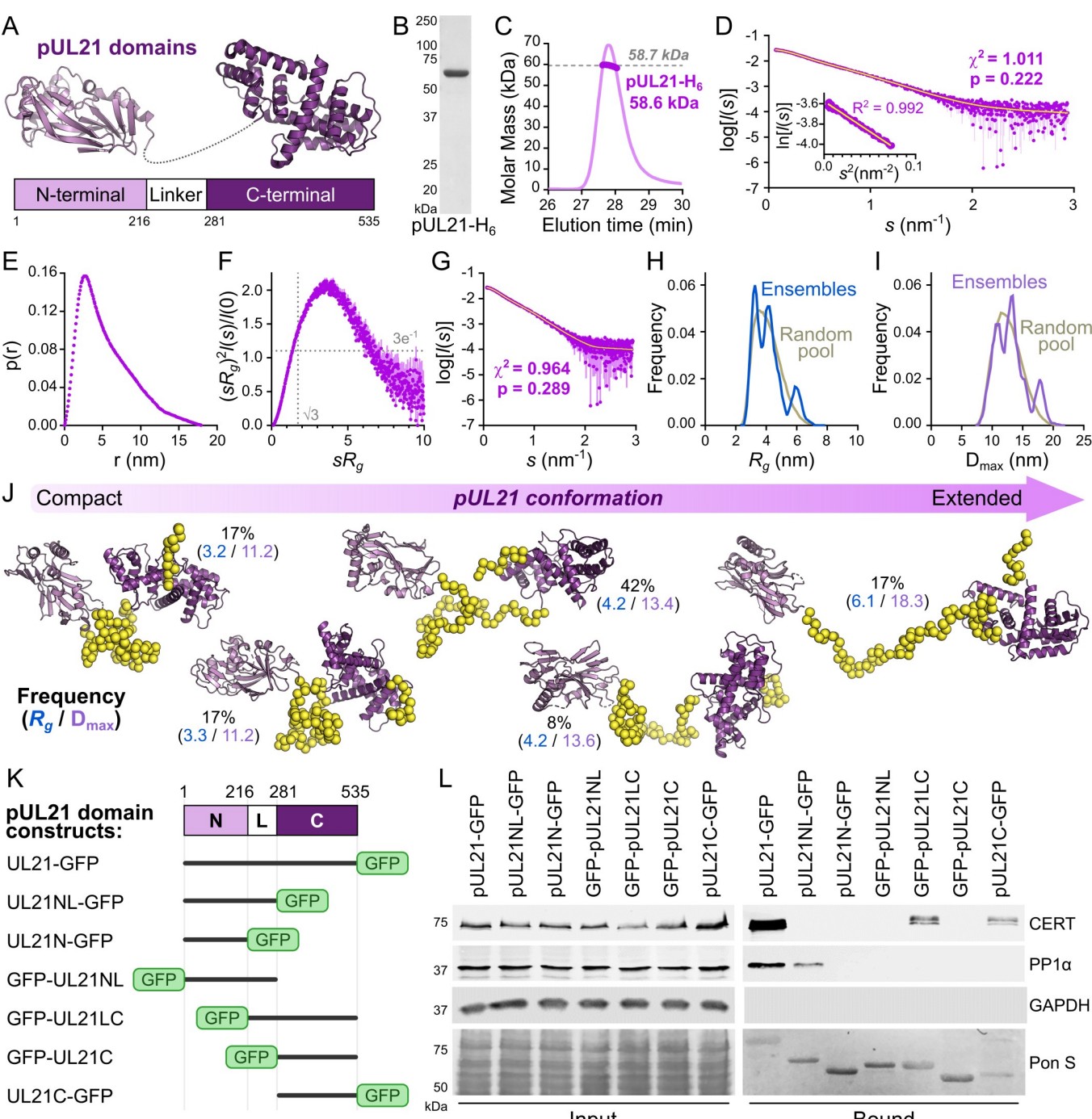

**Fig 2. The C-terminal domain of pUL21 binds CERT whereas the N-terminal domain and disordered linker bind PP1.** (**A**) Schematic representation of pUL21, with crystal structures of the N- and C-terminal domains [24,25] shown in light and dark purple, respectively. (**B**) Coomassie-stained SDS-PAGE of full-length pUL21-$H_6$ purified following bacterial expression. (**C**) SEC-MALS of purified pUL21-$H_6$. Weight-averaged molar mass (thick line) is shown across the SEC elution profile (normalised differential refractive index, thin lines) with the expected molar mass for monomeric pUL21-$H_6$ shown as a dashed line. (**D**) SAXS profile (purple) measured from pUL21-$H_6$ and the corresponding Guinier plot ($sR_g < 1.15$; inset). The Guinier plot is linear (yellow line), as expected for an aggregate-free system. The reciprocal-space fit of the $p(r)$ profile to the SAXS data is shown a yellow trace. $\chi^2$, fit quality; p, Correlation Map (CorMap) probability of systematic deviations between the model fit and the scattering data [93]. (**E**) The real-space distance distribution function, $p(r)$, computed from the experimental SAXS profile. (**F**) Dimensionless Kratky plot of the SAXS data. The expected maximum of the plot for a compact protein is shown as grey dotted lines ($sR_g = \sqrt{3}$, $(sR_g)^2 I(s)/I(0) = 3e^{-1}$). (**G**) Fit to the SAXS profile of a refined pUL21-$H_6$ ensemble obtained by EOM. The ensemble comprises conformational states of the pUL21 N- and C-terminal domains [24,25] joined by a flexible linker. (**H, I**) Comparison of the frequency distributions of $R_g$ (**H**) and $D_{max}$ (**I**) from an initially generated random pool of structures (grey) and the refined EOM ensemble (blue and violet, respectively) indicate that, in

solution, pUL21 samples a distribution of states that encompasses compact, intermediate and extended conformations. (**J**) Selected representative models of the pUL21-H$_6$ ensemble. For each, the prevalence (volume fraction, %), $R_g$ and D$_{max}$ is shown. (**K**) Schematic representation of truncated pUL21 constructs used for immunoprecipitation experiments. (**L**) Immunoblots following immunoprecipitation from HEK293T cells transfected with GFP-tagged full-length pUL21 or truncations thereof. Cells were lysed 24 h post-transfection and incubated with anti-GFP resin to capture protein complexes before being subjected to SDS-PAGE and immunoblotting using the antibodies shown. Pon S, Ponceau S staining of the immunoblot membrane before blocking to show all proteins transferred.

residues near the beginning of the C-terminal domain are required for CERT binding and that binding is blocked by fusion of GFP to the start of this domain.

Mapping the sequence conservation of pUL21 across the family *Alphaherpesvirinae* identifies that the N- and C-terminal domains have more highly conserved sequences than the flexible linker (Fig 3A). However, there is a short stretch of conserved amino acids in the linker region between HSV-1 residues 239–248 [25], with the consensus sequence ϕSxFVQ[VI][KR]xI where ϕ is a hydrophobic residue and x is any residue. As the flexible linker region is required for PP1 binding we hypothesised that these conserved residues may contribute to the interaction. Site-directed mutagenesis was used to generate pUL21-GFP constructs where the conserved hydrophobic residues F242 and V243 were substituted with charged residues glutamate and aspartic acid, respectively, or a double mutant where both residues were substituted for alanine. Immunoprecipitation following transient expression in HEK293T cells showed that these pUL21 mutants retain the ability to bind CERT but no longer bind PP1 (Fig 3B). Immunoprecipitation following ΔpUL21 HSV-1 infection of HEK293T cells transiently expressing pUL21-GFP confirmed that the pUL21$^{FV242AA}$ double-mutant retains the ability to bind pUL16 (Fig 3C). Furthermore, differential scanning fluorimetry (a.k.a. Thermofluor) confirmed that wild-type and FV242AA pUL21-H$_6$ purified following recombinant expression in *E. coli* have similar melting temperatures (Fig 3D). Taken together, these results confirm that substitution of conserved residues in the flexible linker specifically disrupt the ability of pUL21 to recruit PP1 but do not interfere with the overall stability of the protein or its ability to interact with other cellular or viral partner proteins.

*In vitro* dephosphorylation assays with all-purified reagents were used to directly test the hypothesis that pUL21 is a novel viral PP1 regulatory subunit, promoting dephosphorylation of CERT by directly recruiting PP1 catalytic activity. CERT purified from mammalian suspension cell culture is predominantly hyperphosphorylated (CERT$^P$, S2 Fig). The PP1γ catalytic subunit, purified from *E. coli* as a GST fusion to enhance protein solubility, is capable of dephosphorylating CERT when present at high concentrations (Fig 3E and 3F), consistent with low intrinsic specificity of PP1 catalytic subunits [26]. Addition of purified wild-type pUL21-H$_6$ dramatically lowers the concentration of GST-PP1γ required for efficient CERT$^P$ dephosphorylation but this effect is much less pronounced for pUL21$^{FV242AA}$-H$_6$ (Fig 3E and 3F), consistent with specific pUL21-mediated recruitment of CERT$^P$ to PP1. Furthermore, addition of pUL21 did not enhance the dephosphorylation of an irrelevant substrate, phosphorylated eukaryotic initiation factor 2α (eIF2α, Fig 3G), confirming that pUL21 promotes substrate-specific enhancement of PP1 activity rather than acting as an allosteric activator of the phosphatase.

The conservation of the hydrophobic region within the pUL21 linker, which is required for PP1 recruitment, suggested that pUL21 homologues from other alphaherpesviruses may also bind PP1. Immunoprecipitation experiments confirmed that pORF38, the pUL21 homologue from varicella-zoster virus (VZV), also binds PP1 (Fig 3H). As with pUL21, mutation of the conserved hydrophobic residues in pORF38 to alanine (FV255AA) abolished the ability to co-precipitate PP1. Interestingly, neither wild-type nor mutant pORF38 co-immunoprecipitated with CERT, suggesting that the ability to bind and promote CERT dephosphorylation is

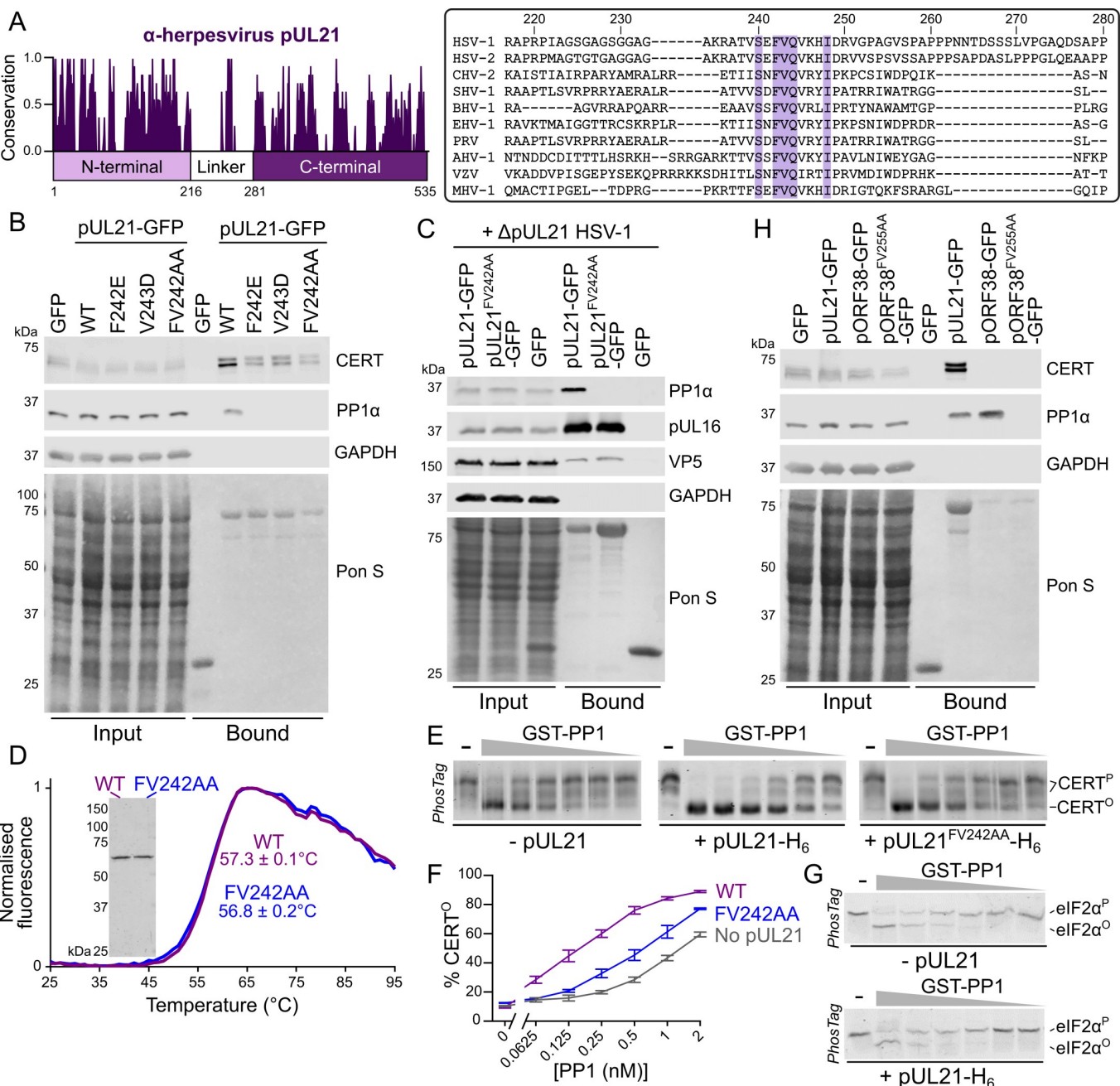

**Fig 3. pUL21 recruits PP1 via a conserved motif in the linker region to accelerate CERT dephosphorylation.** (**A**). Conservation of pUL21 across *Alphaherpesvirinae*. The following sequences were aligned using ClustalW and conservation calculated using Jalview (Abbreviation and Uniprot ID are shown in parentheses): HSV-1 (HSV1, P10205), HSV-2 (HSV2, G9I242), cercopithecine herpesvirus 2 (CHV2, Q5Y0T2), saimiriine herpesvirus 1 (SHV1, E2IUE9), bovine alphaherpesvirus 1 (BHV1, Q65563), equine herpesvirus 1 (EHV1, P28972), pseudorabies virus (PRV, Q04532), anatid herpesvirus 1 (AHV1, A4GRJ2), varicella-zoster virus (VZV, Q6QCT9), turkey herpesvirus (MHV1, Q9DPR5). Alignment across the linker region (residues 217–280 of HSV-1 pUL21) is shown with conserved residues highlighted. (**B**) HEK293T cells were transfected with plasmids expressing GFP, wild-type (WT) pUL21-GFP or pUL21-GFP with amino acid substitutions in the conserved motif. At 24 hours post-transfection the cells were lysed, subjected to immunoprecipitation using a GFP affinity resin, and captured proteins were subjected to SDS-PAGE and immunoblotting using the listed antibodies. Ponceau S (Pon S) staining of the nitrocellulose membrane before blocking is shown, confirming efficient capture of GFP-tagged proteins. (**C**) Plasmids expressing wild-type or mutant pUL21-GFP, or GFP alone, were transfected into HEK293T cells. At 24 hours post-transfection cells were infected with ΔpUL21 HSV-1 (MOI = 5). Cells were lysed 16 hours post-infection and subjected to immunoprecipitation, SDS-PAGE and immunoblotting as in (**B**). (**D**) Differential scanning fluorimetry of WT (purple) and FV242AA substituted (blue) pUL21-H$_6$. Representative curves are shown. Melting temperatures ($T_m$) is mean ± standard deviation (n = 3). Inset shows Coomassie-stained SDS-PAGE of the purified protein samples. (**E**) *In vitro* dephosphorylation assays using all-purified reagents. 0.5 μM CERT was incubated with varying concentrations of GST-PP1 (two-fold serial dilution from 100–3.1 nM) in the absence or presence of 2 μM pUL21-H$_6$ (WT or FV242AA) for 30 min at 30°C. Proteins were

resolved using SDS-PAGE where PhosTag reagent was added to enhance separation of CERT that is hyper- (CERT$^P$) or hypo-phosphorylated (CERT$^O$) and gels were stained with Coomassie. Images are representative of three independent experiments. (**F**) Quantitation of pUL21-mediated stimulation of CERT dephosphorylation, as determined by densitometry. Ratio of CERT$^O$ to total CERT (CERT$^O$ + CERT$^P$) for three independent experiments is shown (mean ± SEM). (**G**) 0.5 μM phosphorylated eIF2α (eIF2α$^P$) was subjected to *in vitro* dephosphorylation using varying concentrations of GST-PP1 (two-fold serial dilution from 200–6.3 nM) in the absence or presence of 2 μM pUL21-H$_6$ as in (**E**). pUL21 does not enhance PP1-mediated dephosphorylation of eIF2α. (**H**) HEK293T cells were transfected with GFP, pUL21-GFP, the VZV homologue of pUL21 with a C-terminal GFP tag (pORF38-GFP), or with pORF38-GFP where amino acid in the conserved motif had been substituted with alanine. Cells were lysed at 24 hours post-transfection and subjected to IP, SDS-PAGE and immunoblotting as in (**B**).

specific to HSV-1 and that pUL21 and its homologues may have additional, conserved targets of PP1-mediated dephosphorylation.

To probe the specific role of PP1 binding during infection, mutant strains of HSV-1 were generated using two-step Red recombination [37] where the conserved residues F242 and V243 in pUL21 were mutated, either individually or in combination (pUL21$^{F242E}$, pUL21$^{V243D}$ and pUL21$^{FV242AA}$). For all mutants, initial virus stocks obtained following transfection of HSV-1 encoding bacterial artificial chromosome into Vero cells (P0 viruses) had impaired replication and/or spread when compared to the wild-type virus, yielding small viral plaques when infecting Vero or HaCaT cells (Figs 4A and S3). This defect was similar in magnitude to that seen when pUL21 expression was completely abolished (ΔpUL21) and the defect was rescued when plaque assays were performed on HaCaT cells constitutively expressing pUL21 (HaCaT pUL21 cells, S3 Fig), confirming that the defect did not arise from other mutations in the HSV-1 genome during virus generation. However, when these pUL21 mutant virus stocks were propagated on Vero cells for three additional generations we observed a rapid rescue of plaque size (Fig 4A): within one generation (P1) a mix of small and large plaques can be observed and by generation three (P3) the mutant virus plaques were as large as those formed by the wild-type virus (Figs 4A and S3). Sanger sequencing of the pUL21 gene amplified by PCR from P3 virus stocks of the three mutant strains revealed that the UL21 gene had not reverted to the wild-type sequence (Fig 4B). Furthermore, analysis of CERT phosphorylation following infection of Vero cells with the P3 mutant viruses confirmed that none of the three mutants had re-acquired the ability to promote CERT dephosphorylation (Fig 4C), strongly suggesting that the ability of pUL21 in these viruses to bind PP1 and recruit it to substrates like CERT remained impaired.

As the rapid increase in spread of these mutant viruses could not be ascribed to reversion of the pUL21 mutation, we hypothesised that the gain of function arose from mutations elsewhere in the HSV-1 genome. Genomic DNA was extracted from the adapted (P3) mutant viruses, and from wild-type and ΔpUL21 viruses subjected to the same propagation strategy in Vero cells, and the genomes were sequenced by next-generation (Illumina) sequencing. Reads were mapped to an HSV-1 strain KOS genome [38] that had been updated to contain all the majority variants present in the wild-type sample, excluding all but one copy of the repeat regions from the analysis. This yielded 115,096 (WT), 248,145 (ΔpUL21), 9,858 (pUL21$^{F242E}$), 46,399 (pUL21$^{V243D}$) and 38,787 (pUL21$^{FV242AA}$) read pairs aligned with high quality, corresponding to 64.7–89.4% coverage of the HSV-1 genome and 71.2–93.9% coverage of the protein coding sequences to a depth of at least ten reads (S4 Fig). Analysis of coding sequence variants between the WT genome and the point mutants identified a striking preponderance of high frequency missense mutations in the US3 open reading frame that encodes the serine/threonine kinase pUS3 (Fig 4D). The US3 variants most prevalent in the pUL21 point mutant viruses (with corresponding amino acid substitutions and frequency) were 135,698 G>T (D207Y; 34.9% in pUL21$^{V243D}$), 135,702 G>A (S208N; 16.2% in pUL21$^{V243D}$, 10.6% in pUL21$^{F242E}$ and 59.2% in pUL21$^{FV242AA}$), 136,127 G>A (A350T; 67.1% in pUL21$^{F242E}$) and

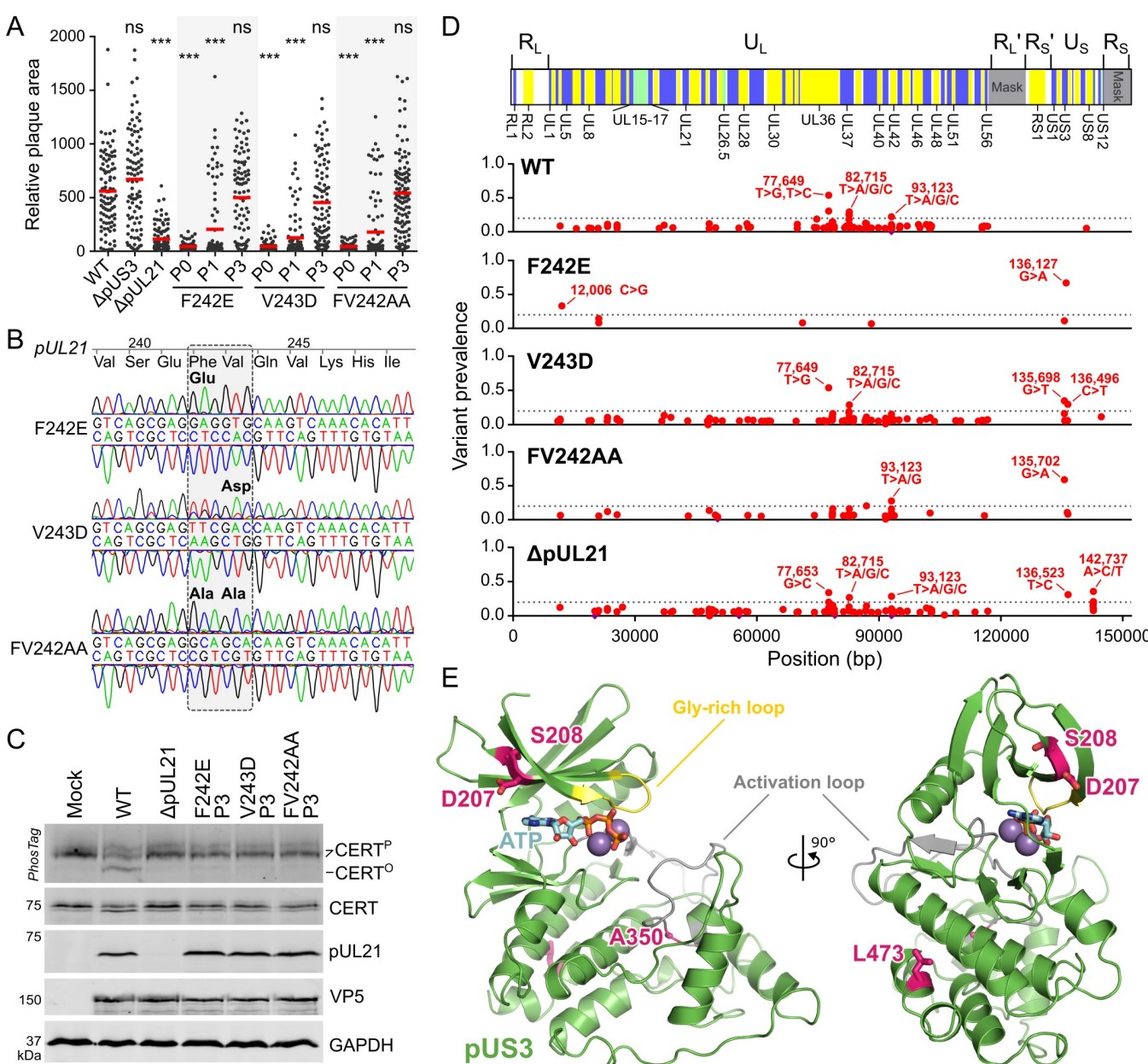

**Fig 4. Mutant HSV-1 where PP1 binding of pUL21 is abolished adapts rapidly to cell culture via compensatory mutations in the kinase gene US3. (A)**. Monolayers of Vero cells were infected with 100 pfu of wild-type (WT), ΔpUS3 and ΔpUL21 HSV-1 or with pUL21 point mutant viruses, harvested immediately following transfection of the recombinant BAC into Vero cells (P0) or following amplification for one (P1) or three (P3) generations in Vero cells. Infected cells were overlaid with medium containing 0.6% carboxymethyl cellulose and incubated for 48 h before fixing and immunostaining with chromogenic detection. Relative plaque areas (pixels) were measured using Fiji. Mean plaque sizes (red bars) were compared to WT using one-way ANOVA with Dunnett's multiple comparisons test (n = 96–113; ns, non-significant; *, $P > 0.05$; **, $P < 0.01$; ***, $P < 0.001$) and images used for quantitation are shown in S3 Fig. (**B**) Sanger sequencing of pUL21 region, amplified by PCR from P3 stocks of virus, show that the introduced point mutations have not reverted to the wild-type sequence. (**C**) Vero cells were mock-infected or infected at MOI = 5 with WT HSV-1, ΔpUL21 virus or with P3 stocks of pUL21 point mutants shown. Lysates were harvested at 16 hpi and subjected to SDS-PAGE plus immunoblotting using the antibodies listed. The upper strip depicts SDS-PAGE where PhosTag reagent was added, retarding the migration of hyperphosphorylated CERT (CERT$^P$) versus the hypophosphorylated protein (CERT$^O$). The P3 stocks of all pUL21 point mutant viruses retain pUL21 expression but all lack the ability to promote CERT dephosphorylation, indicating that PP1 binding has not been restored. (**D**) Prevalence of sequence variants in P3 stocks of pUL21 point mutants and ΔpUL21 when compared to similarly passaged WT HSV-1 as assessed by next-generation sequencing. *Top*: Schematic representation of the HSV-1 genome is shown with alternating background colouring corresponding to HSV-1 genes, green denoting overlapping reading frames, and with selected genes labelled. Repeat regions excluded from the mapping of sequences (*Mask*) are shown in grey. $U_L$ and $U_S$ denote the unique long and unique short segments, respectively, $R_L/R_L'$ are the inverted repeats bounding $U_L$ and $R_S/R_S'$ are the inverted repeats bounding $U_S$. *Bottom*: Non-synonymous variants (red) and insertions/deletions (purple) are shown across the HSV-1 coding sequences. Dotted grey line denotes 20% prevalence and selected high-prevalence variants are labelled. Mutations introduced into pUL21 by

two-step Red recombination are not shown. (**E**) Model of the core kinase domain of pUS3 (residues 189–481) shown in cartoon representation with the catalytically important activation segment (grey) and glycine-rich loop (yellow) highlighted. Amino acid substitutions that are prevalent in adapted pUS21 point mutant viruses are shown as sticks (carbon atoms pink). The position of the catalytic metal ions (magenta spheres) and ATP (sticks, carbon atoms cyan) were modelled by superposition of pUS3 onto the structure of $Mn^{2+}$ and ATP-bound protein kinase A [111].

136,496 C>T (L473F; 29.7% in pUL21$^{V243D}$). Mapping these mutations onto a structural model of the core pUS3 kinase domain (Fig 4E) generated using trRosetta [39] identified that these mutations lie in strand β2 overlaying the ATP binding pocket (D207Y and S208N), within the highly conserved AlaProGlu (APE) motif that forms part of and stabilises the kinase activation segment (A350T), and in helix αI of the C-terminal lobe (L473) that interacts with the APE motif to stabilise the activation loop [40,41]. These substitutions would all be likely to disrupt local folding of pUS3 and thereby alter its activity and/or stability. The ΔpUL21 P3 genome sequence also contained variants in US3, although at lower frequency than in the pUL21 point mutant viruses. The most prevalent was 136,523 T>C (31.2% prevalence), which would abolish the pUS3 stop codon and thereby append an additional 70 amino acids to the expressed protein. A number of low frequency missense mutations were observed in US8A (and the US8/US8A overlap region) in the ΔpUL21 P3 genome sequence. While poor read depth precluded analysis of this region in the pUL21$^{F242E}$ and pUL21$^{V243D}$ viruses, these mutations were not prevalent in the pUL21$^{FV242AA}$ P3 genome (Fig 4D). The missense mutation 12,006 C>G (L122F, 33.3%) present in pUL21$^{F242E}$ would be unlikely to affect the function of pUL4 as a phenylalanine is observed at this position in other alphaherpesviruses. Similarly, the missense mutations observed in pUL36 (77,646 A>C/T/G; 77,649 T>G/C), pUL37 (82,715 T>A/G/C) and pUL42 (93,123 T>A/G/C) are unlikely to have arisen as adaptation to disruption of pUL21 activity as these variants are also observed in the wild-type virus (Fig 4D).

The observation that the US3 kinase gene rapidly mutates to compensate for the inability of pUL21 mutants to recruit PP1 suggests that the two proteins have antagonistic activities, with pUL21 promoting PP1-mediated dephosphorylation of proteins that are phosphorylated by pUS3. The plaques formed by HSV-1 lacking pUS3 expression (ΔpUS3) are indistinguishable from wild-type plaques in Vero cells (Fig 4A). In HaCaT cells the size of ΔpUS3 plaques is greatly reduced (S3 Fig), possibly due to the role of pUS3 in evading antiviral responses [42,43] that are more robust in HaCaT than Vero cells [44,45]. Complementation of pUL21 in *trans* in HaCaT cells restores ΔpUL21 HSV-1 plaques to the size of the wild-type virus (S3 Fig). New stocks of wild-type, pUL21$^{FV242AA}$ and ΔpUL21 HSV-1 were thus prepared via amplification of the original BAC-derived (P0) virus for two generations in HaCaT pUL21 cells (H2 virus), on the basis that pUL21 *trans*-complementation and the selective pressure to maintain pUS3 function in HaCaT cells would synergise to delay the compensatory pUS3 adaptation. The replication of these viruses was assessed using a single-step growth assay where Vero and HaCaT cells were infected at a high multiplicity of infection (MOI) and the production of infectious progeny was measured across the 24 hour replication cycle of the virus (Fig 5A). There is no difference in the growth of wild-type HSV-1 propagated in Vero cell (P4) or HaCaT pUL21 cells (H2). The growth of both pUL21$^{FV242AA}$ H2 and ΔpUL21 H2 HSV-1 was approximately 10 to 100-fold reduced compared to wild-type virus. Furthermore, the plaques formed on Vero cells by the pUL21$^{FV242AA}$ H2 and ΔpUL21 H2 viruses were extremely small (Fig 5B), confirming that these viruses had not adapted to culture when propagated in HaCaT pUL21 cells. The similar growth defect of these two mutant viruses confirms that the ability to recruit PP1 is a critical function of pUL21.

While the unadapted pUL21$^{FV242AA}$ mutant virus exhibited small plaques and reduced growth, the adapted pUL21$^{FV242AA}$ P3 virus has wild-type levels of growth in Vero and HaCaT

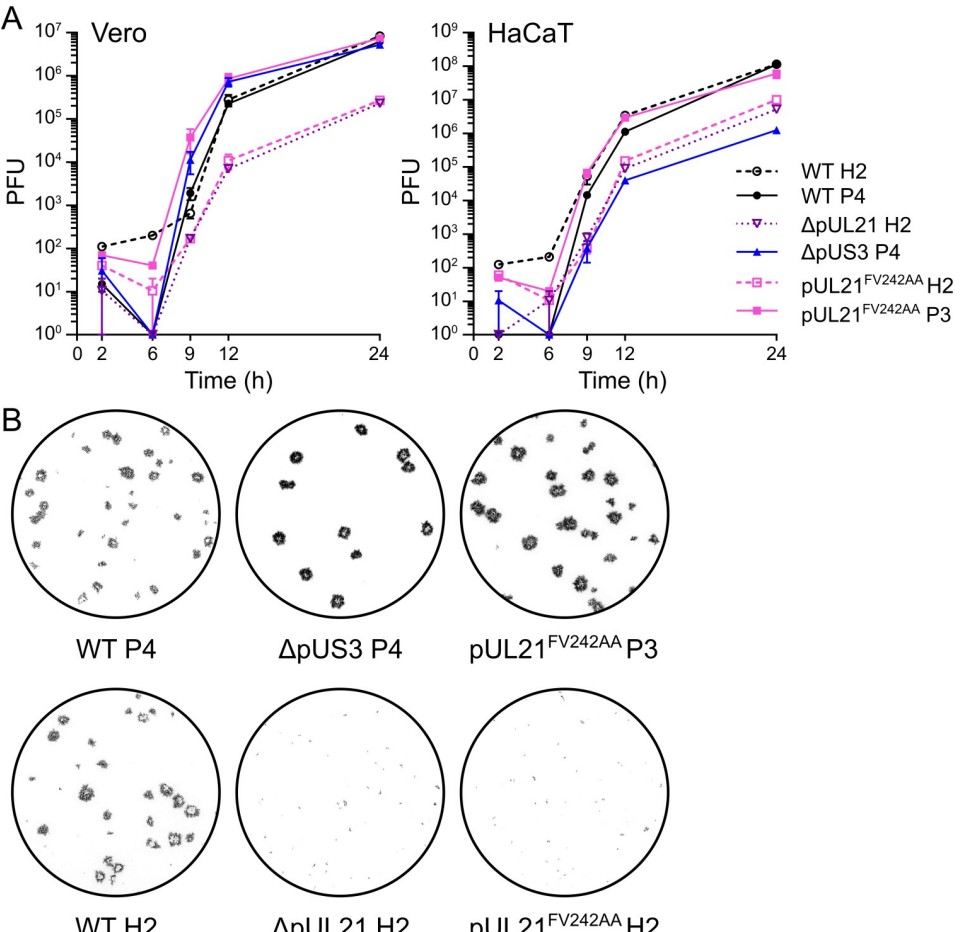

**Fig 5. Abolition of pUL21 PP1 binding severely restricts HSV-1 replication and spread.** (**A**). Single-step (high MOI) growth curve of WT and mutant HSV-1. Monolayers of Vero and HaCaT cells were infected (MOI = 5) with the indicated viruses, prepared following amplification of the original rescued stock for three or four generations in Vero cells (P3 and P4, respectively) or for two generations in HaCaT cells stably expressing pUL21 (H2). Samples were harvested at the indicated times and titers were determined by plaque assay using Vero cells. Data are presented as mean values of duplicates of one representative experiment. Error bars represent standard error of the mean (not shown where errors are smaller than the symbols). (**B**) Representative plaque assays used to titre the single-step growth curve shown in (**A**). Monolayers of Vero cells were infected with viruses harvested from titration, overlaid with medium containing 0.6% carboxymethyl cellulose, and then fixed and immunostained with chromogenic detection at 48 h post-infection.

cells (Fig 5A). This restoration of growth to wild-type levels is particularly striking because the replication of the pUL21^{FV242AA} P3 virus on HaCaT cells dramatically exceeds that of the ΔpUS3 virus (Fig 5A). This suggests that abolishing kinase (ΔpUS3) or phosphatase (ΔpUL21 and non-adapted pUL21^{FV242AA}) activity individually is more deleterious to virus replication than simultaneously modulating both counteracting activities.

The HSV-1 pUS3 kinase has a broad substrate specificity [46] that overlaps with those of the cellular kinases Akt [47] and protein kinase A (PKA) [48]. Immunoblotting confirmed an increase in phosphorylated Akt and PKA substrates when Vero cells were infected with wild-type HSV-1 but not when infected with ΔpUS3 HSV–1 (Fig 6A). The abundance of several phosphorylated substrates is further increased in cells infected with ΔpUL21 or unadapted (H2) pUL21^{FV242AA} HSV-1, whereas the adapted pUL21^{FV242AA} P3 virus exhibits an overall reduction in the phosphorylation of the pUS3 targets (Fig 6A). The same changes were also

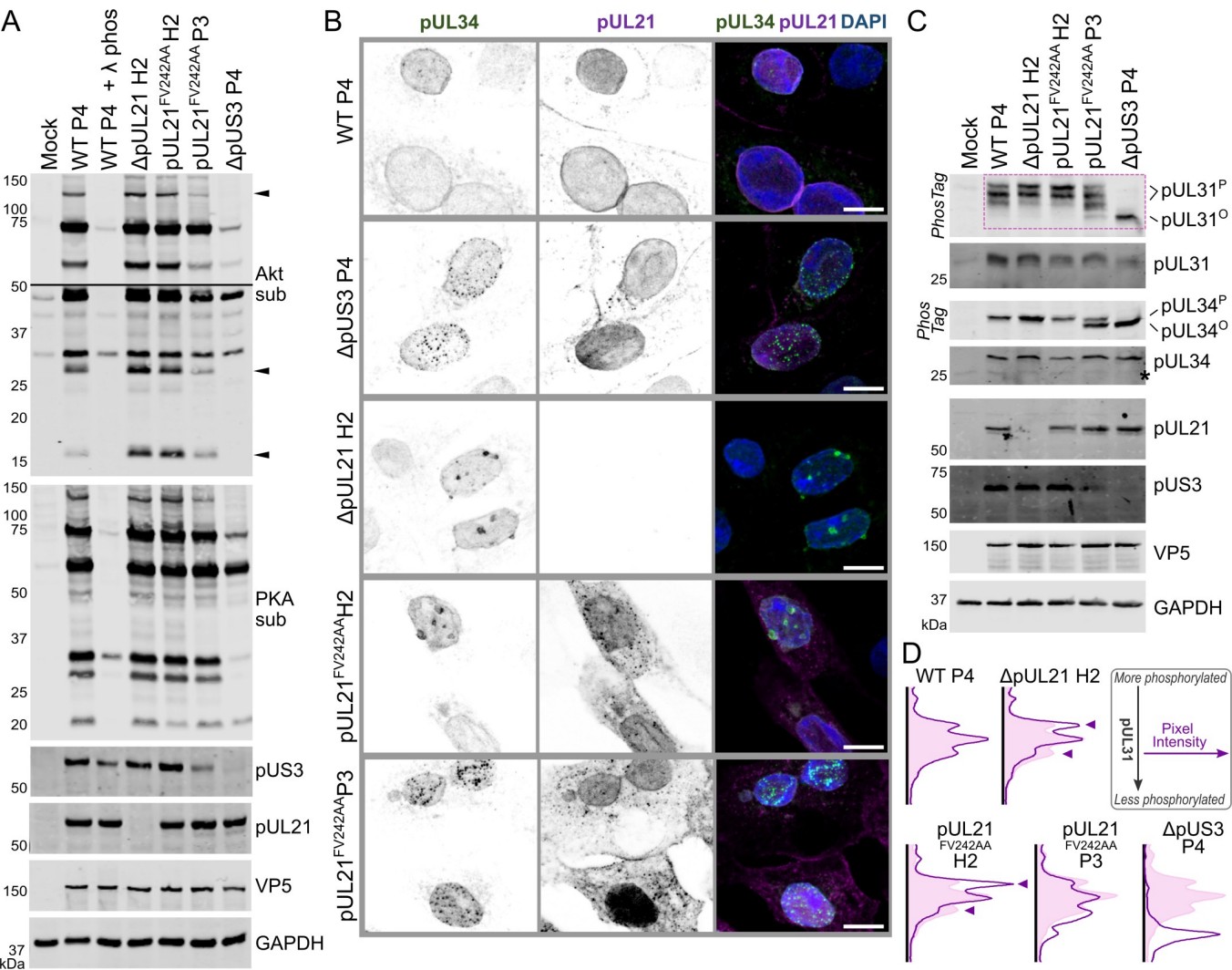

**Fig 6. pUL21 antagonises pUS3-mediated protein phosphorylation and NEC distribution.** (**A**). Vero cells were infected at MOI = 5 with wild-type (WT) or mutant HSV-1 that had been prepared following amplification of the original rescued stock for three or four generations in Vero cells (P3 and P4, respectively) or for two generations in HaCaT pUL21 cells (H2). Lysates were harvested at 16 hpi and subjected to SDS-PAGE plus immunoblotting. With the exception of the lysate from WT HSV-1 infected cells that was treated with lambda phosphatase (λ phos), all lysates were harvested in the presence of phosphatase inhibitors. Immunoblots were probed with the antibodies shown. Antibodies recognising phosphorylated Akt and PKA substrates (sub) illustrate the activity of pUS3, as the specificity of this viral kinase overlaps with those of cellular kinases Akt and PKA [47,48]. For the Akt sub immunoblot two exposures are shown, separated by a line, and phosphorylated Akt substrates more abundant in cells infected with virus lacking pUL21 that can recruit PP1 (ΔpUL21 and pUL21^FV242AA H2) are marked with arrowheads. (**B**) Vero cells were infected at MOI = 1 with WT or mutant HSV-1, prepared as in (**A**). Cells were fixed at 10 hpi and stained using antibodies that recognise pUL34 (green) and pUL21 (magenta). The merge includes DAPI (blue) and the scale bar represents 10 μm. (**C**) Vero cells were infected at MOI = 5 with WT or mutant HSV-1 as listed. Lysates were harvested at 16 hpi and subjected to SDS-PAGE plus immunoblotting using the antibodies listed. The upper strips of the pUL31 and pUL34 blots depict SDS-PAGE where PhosTag reagent was added to enhance separation of hyperphosphorylated (pUL31^P and pUL34^P) and hypophosphorylated (pUL31^O and pUL34^O) forms of the proteins. Non-specific bands are indicated with an asterisk (*). Boxed region denotes pUL31 bands used for quantitation in (**D**). (**D**) Relative abundance of pUL31 phosphoforms. Two-dimensional intensity profiles of pUL31 immunoblots following PhosTag SDS-PAGE (**C**), calculated by averaging horizontal pixel intensities for each lane (arbitrary units) along the vertical axis, are shown as purple curves. The intensity profile for pUL21 WT P4 infected cells is shown for comparison as a pink filled curve. Prominent differences in the profiles for cells infected with virus lacking pUL21 that can recruit PP1 (ΔpUL21 and pUL21^FV242AA H2) are denoted with arrowheads.

observed in HaCaT cells infected with the same panel of viruses (S5A Fig). This confirms that the mutations observed in the adapted virus impair pUS3 kinase activity and suggests that pUL21 may directly antagonise the pUS3 activity for some substrates.

One of the best characterised roles of pUS3 is in the regulation of herpesvirus nuclear egress. pUS3 helps reorganize the nuclear lamina by phosphorylating lamin A/C [46], promotes redistribution of the inner nuclear membrane associated protein emerin [49,50], and may facilitate fusion of the primary envelope with the outer nuclear membrane via phosphorylation of gB [51]. Furthermore, pUS3 has been shown to directly phosphorylate the NEC components pUL31 [52,53] and pUL34 [54,55] and thereby regulate nuclear egress of capsids. Recent evidence suggests that pUL21 may also regulate NEC function as nuclear egress is severely disrupted in HSV-2 where pUL21 expression is absent [12], and in both HSV-1 and HSV-2 infection the distribution of the NEC in the nuclear membrane was perturbed when either pUS3 or pUL21 expression was abolished [18]. Immunocytochemistry showed a smooth distribution of the NEC component pUL34 around the nuclear rim in wild-type HSV-1 infected Vero cells (Fig 6B). pUL21 partially co-localises with pUL34 at the nuclear rim, also being observed in the nucleoplasm, at cytoplasmic punctae and at the cell surface (Fig 6B). As previously identified [18], we observed pUL34 distributed as small punctae on the nuclear rim in cells infected with ΔpUS3 HSV-1 whereas pUL34 formed large punctae in cells infected with ΔpUL21 virus (Fig 6B). The distribution of pUL34 in cells infected with unadapted pUL21$^{FV242AA}$ H2 HSV-1 closely resembles the ΔpUL21 infection, with large punctate accumulations of pUL34 observed at the nuclei of infected cells (Fig 6B). The subcellular localisation of pUL21 was broadly similar between this mutant and the wild-type virus, confirming that the conserved motif is not required for recruitment of pUL21 to the nuclear rim, although the pUL21 cytoplasmic punctae are more prominent (Fig 6B). Strikingly, the distribution of pUL34 in cells infected with the adapted pUL21$^{FV242AA}$ P3 mutant resembles that of ΔpUS3 virus (Fig 6B), showing that the small-punctae phenotype arising from altered pUS3 kinase activity [53] is dominant over the large-punctae phenotype that may arise from an absence of pUL21-directed phosphatase activity. Similar changes to pUL34 localisation were also observed in HaCaT cells infected with the same panel of viruses (S5B Fig).

To directly assess the effects upon NEC phosphorylation, the level of phosphorylated pUL31 and pUL34 was assessed following infection of Vero cells. As observed previously [52,53,55], pUL31 and pUL34 are phosphorylated in cells infected with wild-type HSV-1 but not in cells infected with a ΔpUS3 mutant (Fig 6C). The extent of pUL31 hyperphosphorylation is increased when cells are infected with ΔpUL21 HSV-1 or the unadapted pUL21$^{FV242AA}$ H2 virus (Fig 6C and 6D). However, the extent of pUL34 phosphorylation remains unaltered, consistent with pUL21 specifically targeting pUL31 but not pUL34 for PP1-mediated dephosphorylation. The extent of pUL31 phosphorylation in cells infected with the adapted pUL21$^{FV242AA}$ P3 mutant virus is similar to wild-type infection and pUL34 phosphorylation is reduced compared to wild-type, consistent with impaired pUS3 kinase activity in the adapted virus. The abundance of pUS3 is decreased in pUL21$^{FV242AA}$ P3 infected cells compared with wild-type HSV-1 or other pUL21 mutants (Fig 6A and 6C), in accordance with the adaptive mutations reducing pUS3 stability in addition to altering its kinase activity. Similar changes to the phosphorylation of pUL31 and pUL34 are observed in infected HaCaT cells (S5C Fig). However, we note reduced abundance of viral proteins in HaCaT cells infected with ΔpUS3 HSV-1, presumably due to restricted virus infection caused by an absence of pUS3-mediated innate immune evasion [42,43]. The abundance of pUL31 is particularly diminished in the ΔpUS3 infection, suggesting that pUS3 may be required for pUL31 stability in HaCaT cells (S5C Fig).

## Discussion

The HSV-1 tegument protein pUL21 and its homologues across herpesviruses have been implicated in multiple functions relating to virus replication and spread [12,18–21] but a molecular mechanism had remained elusive. We show here that pUL21 is a PP1 adaptor that regulates the phosphorylation of specific cellular and viral proteins, thereby promoting virus replication and spread. In particular, pUL21 acts to counterbalance the kinase activity of pUS3. The roles of phosphatases in cell biology are understudied when compared to their kinase brethren [56] and our identification of pUL21 as a viral regulator of PP1 activity highlights a novel layer of infected-cell protein phosphorylation control by herpesviruses.

The importance of pUL21 phosphatase adaptor activity for HSV-1 replication and spread is highlighted by the rapid adaptation via mutation of the viral kinase pUS3 when the ability of pUL21 to bind PP1 is impaired (Fig 4). This is strong genetic evidence that pUL21-mediated phosphatase activity directly counteracts the kinase activity of pUS3, although we note that sequencing of adapted populations rather than individual plaque-purified isolates may confound the identification of mutations elsewhere in the genome that synergise with US3 mutations to promote replication of pUL21 PP1-binding mutants. The serine/threonine kinase pUS3 is an important alphaherpesvirus virulence factor that phosphorylates numerous viral and cellular targets [57]. Importantly, we observe that pUL21 promotes dephosphorylation of pUS3 substrates, including the NEC component pUL31 (Fig 6). The fact that pUL21 promotes dephosphorylation of pUL31 but not pUL34, both substrates of pUS3, is consistent with pUL21 directly promoting dephosphorylation of selected pUS3 targets rather than acting upon the phosphorylation state, and thus kinase activity, of pUS3 [58]. Immunoblotting for phosphorylated Akt substrates in HSV-1 infected cells reveals additional proteins targeted by pUL21 for PP1-mediated dephosphorylation that have yet to be identified and confirms that pUL21 has a broad specificity that overlaps with, but is not identical to, that of pUS3. In contrast to pUS3, which is expressed immediately after infection and plateaus at around 6 hours post-infection (hpi), pUL21 is a late gene and its abundance increases steadily as the infection progresses [27]. One could thus hypothesise that pUL21 confers temporal regulation of pUS3-dependent protein phosphorylation, promoting dephosphorylation of specific pUS3 substrates at late times post-infection to ensure coordinated progression of virus replication. pUL21 may also provide spatial regulation, for example by associating with nuclear capsids [15,21] and thus promoting pUL31 dephosphorylation once these capsids have docked with the NEC to promote capsid primary envelopment [46]. This is consistent with recent data suggesting pUS3 activity controls NEC self-association and resulting nuclear membrane deformation [59]. Importantly, pUL21 also dephosphorylates distinct cellular targets such as CERT that are not phosphorylated by pUS3; as such pUL21 does not just counterbalance pUS3-driven protein phosphorylation but also actively remodels the phosphoproteomic landscape of infected cells.

Given the rapid adaptation of the pUL21$^{FV242AA}$ virus, where PP1 binding is abolished, it is perhaps surprising that rapid adaptation is not observed in the HSV-1 mutant where pUL21 expression is abolished (ΔpUL21). We did not see outgrowth of syncytial mutants nor purifying selection for mutations in gK (Fig 4), in contrast to a previous report [20]. We do observe some genetic evidence of pUS3 mutations in ΔpUL21 HSV-1 that has been passaged in Vero cells, although not to the same extent as the pUL21 point mutants (Fig 5), and our biochemical studies confirm that pUS3 activity is maintained in the ΔpUL21 virus (Fig 6). These differences in pUS3 adaptation can be ascribed to the multi-functional nature of pUL21. Although ΔpUL21 and pUL21$^{FV242AA}$ HSV-1 have similar growth defects, the ΔpUL21 virus lacks all pUL21-mediated activities whereas pUL21$^{FV242AA}$ lacks only the ability to bind PP1 and

promote protein dephosphorylation. In the context of ΔpUL21 the inactivation of pUS3 would not overcome the lack of pUL21 to bind pUL16, as part of the pUL21-pUL16-pUL11 complex that binds gE and promotes virus spread [19,20], or any additional as-yet unidentified activities of pUL21. As such, pUS3 mutations in the ΔpUL21 virus would be expected to confer less of a growth advantage. While confirmation of this hypothesis awaits identification of pUL21 mutations that abolish its ability to bind pUL16, the difference in behaviour of the ΔpUL21 and pUL21$^{FV242AA}$ viruses highlights the importance of mapping specific protein:protein interactions and identifying amino acids that disrupt these interactions in order to dissect the mechanisms by which multifunctional viral proteins promote virus replication and spread.

In addition to promoting dephosphorylation of NEC component pUL31, we show that pUL21 directly binds the ceramide transport protein CERT and promotes its dephosphorylation (Figs 1 and 3). Activated CERT transfers ceramide from the ER to the TGN where sphingomyelin synthase converts ceramide and phosphatidylcholine to sphingomyelin and diacylglycerol, the latter lipid promoting the recruitment and activation of protein kinase D [29,60]. PKD activity is known to modulate HSV-1 replication/spread [61] and a previous study of PKD modulators showed that siRNA depletion of CERT promoted aberrant secretion of HSV-1 particles [62]. While depletion of CERT is opposite in effect to pUL21-mediated dephosphorylation, which would increase CERT activity, it is tempting to speculate that pUL21-mediated modulation of CERT activity specifically changes the lipid composition of infected cells to the benefit of the virus, either by ensuring the correct lipid composition of virus assembly compartments or by altering post-Golgi protein trafficking by modulating the abundance of sphingolipids [63,64]. Our data shows that CERT hypophosphorylation is not required for HSV-1 replication and spread in cultured epithelial cells, as the adapted pUL21$^{FV242AA}$ P3 mutant HSV-1 has wild-type levels of replication and spread but lacks the ability to dephosphorylate CERT (Figs 4 and 5). The absence of observable growth defect suggests that CERT hypophosphorylation could promote infection in other cell types or have an immunomodulatory role during infection *in vivo*, topics for future study.

Both cellular and viral adaptors of PP1 recruit the catalytic subunits via small linear motifs (SLiMs), the most common being the "RVxF" motif [33,65]. For example, the HSV protein ICP34.5 (a.k.a. γ$_1$34.5) recruits PP1 via an RVxF motif to promote dephosphorylation of eIF2α, thereby alleviating the translational blockade promoted by the infection-stimulated protein kinase R [66,67], measles virus V protein sequesters PP1 via an RVxF-like motif to prevent MDA5 dephosphorylation [68], and recruitment of PP1 via an RvXF-like motif by the HIV Tat protein promotes HIV transcription [69]. While pUL21 can clearly act in isolation as a PP1 adaptor protein to promote CERT dephosphorylation *in vitro* (Fig 3E and 3F), it is unclear whether pUL21 acts synergistically with cellular adaptors or competes with them for PP1 binding in the context of infected cells. pUL21 does not contain any sequence that matches the full RVxF consensus sequence [33]. While pUL21 does contain two sequences that match a less stringent definition of the RVxF motif [34], these are in structured regions of the protein and unlikely to interact with PP1 (S1 Fig) and neither is conserved in pORF38, which also binds PP1. However, we identified that a conserved motif in the highly-flexible pUL21 linker region, with consensus sequence φSxFVQ[VI][KR]xI (where φ is a hydrophobic residue and x is any amino acid), is required for PP1 binding in both pUL21 and the VZV homologue pORF38 (Fig 3). We hypothesise that this Twenty-one Recruitment Of Protein Phosphatase One (TROPPO) motif is a novel PP1 SLiM. While interfering with PP1 activity via broad-spectrum inhibitors is toxic [70], the use of small molecules to block interactions between PP1 and specific SLiMs is a promising avenue for the development of new therapeutics [71]. Indeed, this approach has already been used in the development of compounds that inhibit HIV and Ebola virus replication in cultured cells [72,73]. Our identification of the

TROPPO motif raises the exciting prospect that pUL21 binds PP1 via a non-canonical surface that could be precisely targeted by small molecules with minimal off-target effects, although confirmation of this awaits structural characterisation of PP1 in complex with the TROPPO motif.

In summary we have observed that pUL21 functions as a novel viral phosphatase adaptor, directly recruiting the catalytic subunit of PP1 to promote dephosphorylation of specific cellular and viral phosphoproteins. The PP1-binding motif of pUL21 is absolutely conserved across alphaherpesviruses and mutation of this motif severely restricts virus growth, making the PP1 binding surface for this motif an attractive target for therapeutic design. Compensatory mutations in the viral kinase pUS3 rapidly accumulate when PP1 binding of pUL21 is mutated and these adaptations promote virus growth and spread. This highlights the caution that must be exercised when investigating individual virus kinases and phosphatases as optimal virus growth and spread requires a fine balance of the two and perturbation provides a potent selective pressure for compensatory mutations.

## Materials and methods

### Ethics statement

All procedures for mouse immunizations were approved by the University of Cambridge ethical review board and by the UK Home Office under the 1986 Animal (Scientific Procedures) Act (project license 80/2538).

### Plasmids

For transfection of mammalian cells, HSV-1 strain KOS pUL21 (UniProt F8RG07) or truncations thereof was cloned into pEGFP-N1 (Clontech) for expression with a C-terminal GFP tag or into pEGFP-C2 (Clontech) for expression with an N-terminal GFP tag and a codon-optimised synthetic gene (GeneArt) encoding VZV isolate HJ0 pORF38 (UniProt Q2PJ26) was cloned into pEGFP-N1. For generation of stable cells, pUL21 was cloned into pDONR before being sub-cloned into pLentiCMV/TO PuroDEST [74] using Gateway cloning (Thermo-Fisher) as per the manufacturer's instructions, and lentivirus particles were generated using the psPAX.2 (packaging) and pMD2.G (VSV G envelope) helper plasmids. For purification following bacterial expression, pUL21 was cloned into pOPTnH [75] encoding a C-terminal $KH_6$ tag. For purification following mammalian expression, a codon-optimised synthetic gene (GeneArt) encoding human CERT (UniProt Q9Y5P4-2) was cloned into pcDNA-SW, a derivative of pcDNA3.1(+) encoding an N-terminal StrepII-tag and a woodchuck hepatitis virus posttranscriptional regulatory element [76] in the $3'$ untranslated region of the cloned gene. Site directed mutagenesis of pUL21 and pORF38 was performed using QuikChange mutagenesis (Agilent) according to the manufacturer's instructions. To generate recombinant viruses, pEPkan-S, containing an I-SceI/KanR selection cassette was used [37]. A plasmid (UK622) encoding mouse PP1γ (UniProt P63087) with an N-terminal GST tag [77] was a kind gift from David Ron (Cambridge Institute for Medical Research) and the core catalytic domain of mouse PP1γ (residues 7–300) was sub-cloned into pOPTnH, encoding a C-terminal $KH_6$ tag.

### Mammalian cell culture

Mycoplasma-free human embryonic kidney (HEK)293T cells (ATCC #CRL-3216), spontaneously immortalised human keratinocyte (HaCaT) cells [78] and African green monkey kidney (Vero) cells (ATCC #CRL-1586) were grown in Dulbecco's Modified Eagle Medium with high glucose (Sigma), supplemented with 10% (v/v) heat-inactivated foetal calf serum (FCS) and 2

mM L-glutamine (complete DMEM) in a humidified 5% $CO_2$ atmosphere at 37˚C. For stable isotope labelling of amino acids in cell culture (SILAC) experiments, HEK293T cells were grown in SILAC medium (high glucose DMEM lacking arginine and lysine, Life Technologies) supplemented with 10% (v/v) dialyzed (7 kDa molecular weight cut-off) heat-inactivated FCS, 2 mM glutamine, 100 U/mL penicillin and 100 μg/mL streptomycin. Media were supplemented with 84 mg/L arginine (light, unlabelled; medium, Arg6 ($^{13}C6$); heavy, Arg10 ($^{13}C6$, $^{15}N4$)) and 146 mg/L lysine (light, unlabelled; medium, Lys4 ($^2H4$); heavy, Lys8 ($^{13}C6$, $^{15}N2$)) and all media were 0.2 μm sterile filtered prior to use. Cells were maintained in SILAC media for at least five passages before use to ensure complete labelling. For purifying proteins, Freestyle 293F suspension cells (ThermoFisher) were maintained in Freestyle 293F medium (Gibco) on a shaking platform (125 rpm) in a humidified 8% $CO_2$ atmosphere at 37˚C.

HaCaT cells stably expressing pUL21 (HaCaT-pUL21) were generated by lentivirus transduction. $3 \times 10^5$ HEK293T cells were transfected with a 2:1:2 mass ratio of psPAX.2:pMD2.G: pLentiCMV/TO PuroDEST pUL21 (815 ng total DNA) using Lipofectamine 2000 (ThermoFisher) as per the manufacturer's instructions. The cell supernatant was harvested at 48 and 72 h post-transfection, supernatants were pooled and debris was removed by low-speed centrifugation. $5 \times 10^4$ HaCaT cells were transduced by incubation with a 1:50 dilution of the virus-containing supernatants, or mock transduced, in complete DMEM supplemented with 100 U/mL penicillin, 100 μg/mL streptomycin and 10 μg/mL polybrene for 48 h before successfully transduced cells were selected by culture in complete DMEM supplemented with 100 U/mL penicillin, 100 μg/mL streptomycin and 2 μg/mL puromycin until all cells in the mock transduction had died (72 h). Clones derived from single cells were isolated by cell sorting and expression of pUL21 was confirmed by immunoblot and immunocytochemistry (see below).

## GFP affinity capture and quantitative mass spectrometry

Monolayers of HEK293T cells were transfected with Lipofectamine 2000 (ThermoFisher) or TransIT-LT1 (Mirus) using 7.7 μg of pUL21-GFP or pEGFP-N1 (for GFP alone) per 9 cm dish of cells, in accordance with the manufacturer's instructions, and the relevant labelled medium was used to prepare the transfection reagents. Cells were harvested 24 h post-transfection by scraping into the medium, pelleted (220 g, 5 min, 4˚C) and washed three times with cold PBS. Cells were lysed at 4˚C in 1 mL lysis buffer (10 mM Tris pH 7.5, 150 mM NaCl, 0.5 mM EDTA, 0.5% NP-40, 1:100 diluted EDTA-free protease inhibitor cocktail (Sigma-Aldrich)) for 45 min before clarification (20,000×g, 10 min, 4˚C). Protein concentration in the lysates was quantified by BCA assay (Thermo Scientific) to equalise protein concentrations across the samples before immunoprecipitation with GFP-TRAP beads (ChromoTek) following the manufacturer's protocol, samples being eluted by incubation at 95˚C for 5 min in 45 μL 2× SDS-PAGE loading buffer. Input and bound samples were separated by SDS-PAGE and analyzed by immunoblot. For each biological repeat (n = 3), 8 μL of light-, medium- and heavy-labelled eluted samples were mixed in a 1:1:1 ratio and frozen at -80˚C until mass spectroscopy analysis.

Mass spectrometry analysis was performed by the proteomics facility of the University of Bristol (UK) as described in [27]. The raw data files were processed using MaxQuant v. 1.5.6.0 [79]. The in-built Andromeda search engine [80] was used to search against the human proteome (UniProt UP000005640, canonical and isoform entries, accessed 11/09/2016) and a custom proteome file containing the sequence of HSV-1 strain KOS pUL21. Trypsin/P digestion, standard modifications (oxidation, N-terminal acetylation) were selected as group-specific parameters and SILAC quantification was performed using light (Arg0, Lys0), medium (Arg6, Lys4) and heavy (Arg10, Lys8) labels. Re-quantification, razor protein FDR, and second

peptide options were enabled for the processing. The quantified data were analysed with a custom R script using the normalized ratios obtained by MaxQuant. Proteins only identified by site or against the reverse database, as well as common experimental contaminants such as keratins (specified in the MaxQuant contaminants file) were removed. Only proteins identified in at least two of the three biological repeats (325/464) were considered for analysis. Significance of abundance changes was determined using a one-sample, two-sided t-test. The mass spectrometry proteomic data has been deposited with the ProteomeXchange Consortium (http://www.proteomexchange.org/) via the PRIDE partner repository [81] under the data set identifier PXD027257.

## Antibodies

The following antibodies with listed dilutions were used for immunoblotting: anti-CERT 1:10,000 (Abcam, ab72536), anti-pUL21 1:50 (this study, see below), anti-PP1α 1:1000 (Santa Cruz, sc-271762), anti-PP1β 1:1000 (Santa Cruz, sc-373782), anti-PP1γ 1:1000 (Santa Cruz, sc-6108), anti-pUL34 1:500 [18], anti-pUL31 1:200 [18], anti-pUS3 1:1000 [82], anti-VP5 [DM165] 1:50 [83], mouse anti-GAPDH 1:10,000 (GeneTex, GTX28245), rabbit anti-GAPDH 1:10,000 (GeneTex, GTX100118), anti-pUL16 1:2000 [84], anti-phospho-PKA substrates 1:1000 (Cell Signalling, 9624), anti-phospho-Akt substrates 1:1000 (Cell Signalling, 9611). After incubation with the 1:10,000 dilutions of fluorescently labelled secondary antibodies: LI-COR IRDye 680T conjugated goat anti-rat (926–68029), donkey anti-rabbit (926–68023) or goat anti-mouse (926–68020), or LI-COR IRDye 800CW conjugated donkey anti-rabbit (926–32213), donkey anti-goat (926–32214), donkey anti-chicken (926–32218) or goat anti-mouse (926–32210), the signals were detected using Odyssey CLx Imaging System (LI-COR) and analysed using Image Studio Lite software (LI-COR). Primary antibodies used for immunocytochemistry were anti-pUL21 1:10 (this study, see below), anti-pUL34 1:4000 [85], and secondary antibodies were 1:1000 dilutions of Alexa Fluor 647 conjugated goat anti-mouse (Invitrogen, A21236) and Alexa Fluor 488 conjugated goat anti-chicken (Invitrogen, A11039). The primary antibody used for visualising HSV-1 plaques was anti-gD (LP2) [86] diluted 1:50, and the secondary antibody was HRP-conjugated rabbit anti-mouse (DaKo, P0161) diluted 1:5000.

A monoclonal antibody to pUL21 was generated by inoculating female BALB/c mice with HSV-1 using ear scarification, followed by a virus protein boost 1 month later. After 3 days, spleens were harvested and hybridomas were generated as described previously [87]. Hybridomas secreting antibodies specific to pUL21 were identified by immunofluorescence screening and cloned by limiting dilution. Hybridoma supernatants were 0.2 μm filtered and maintained at 4˚C for short-term storage or -20˚C for long-term storage. The antibody specific to pUL21 used in this work recognizes an epitope contained within the C-terminal domain (amino acids 280–535) of pUL21.

## Recombinant protein purification following bacterial expression

Recombinant proteins were expressed in *Escherichia coli* T7 Express lysY/I$^q$ cells (New England Biolabs). For all proteins except PP1γ, cells were grown in 2×TY medium at 37˚C to an OD600 of 0.8–1.2 before cooling to 22˚C and inducing protein expression by addition of 0.4 mM isopropyl β-D-thiogalactopyranoside (IPTG). Cells were harvested at 16–20 h post-induction by centrifugation and pellets were stored at -70˚C until required. For GST-PP1γ, the 2xTY medium was supplemented with 1 mM MnCl$_2$ and the cultures were cooled to 18˚C upon reaching an OD600 of 0.8, followed by induction using 1 mM IPTG. For PP1γ(7–300)-H$_6$, the 2xTY medium was supplemented with 1 mM MnCl$_2$ and cultures were cooled to 21˚C

after reaching A600 of 0.8, followed by induction using 0.4 mM IPTG. For all recombinant proteins, cells were resuspended in lysis buffer (see below) at 4˚C before lysis using a TS series cell disruptor (Constant Systems) at 24 kpsi. Lysates were cleared by centrifugation (40,000 g, 30 min, 4˚C) and incubated with the relevant affinity resins for 1 h at 4˚C before extensive washing ($\geq$ 20 column volumes) and elution using the relevant elution buffer (see below). Samples were concentrated and subjected to size exclusion chromatography (SEC, see below), fractions containing the desired protein as assessed by SDS-PAGE being pooled, concentrated, snap-frozen in liquid nitrogen and stored at -70˚C.

The lysis buffer for pUL21-GST (20 mM Tris pH 8.5, 300 mM NaCl, 0.5 mM $MgCl_2$, 1.4 mM β-mercaptoethanol, 0.05% TWEEN-20) and GST-PP1γ (50 mM Tris pH 7.5, 500 mM NaCl, 1 mM $MnCl_2$, 0.5 mM $MgCl_2$, 1.4 mM β-mercaptoethanol, 0.05% TWEEN-20) was supplemented with 200–400 U bovine DNase I (Sigma) and 200 μL EDTA-free protease inhibitor cocktail (Sigma). Cleared lysates were incubated with glutathione Sepharose 4B (GE Healthcare), washed with wash buffer (20 mM Tris pH 8.5 [pUL21-GST] or 7.5 [GST-PP1γ], 500 mM NaCl, 1 mM DTT, plus 1 mM $MnCl_2$ [GST-PP1γ only]) and protein eluted using wash buffer supplemented with 25 mM reduced glutathione. SEC was performed using a HiLoad Superdex S200 16/600 column (GE Healthcare) equilibrated in 20 mM Tris pH 8.5, 500 mM NaCl, 1 mM DTT (pUL21-GST) or 50 mM Tris pH 7.5, 100 mM NaCl, 1 mM DTT (GST-PP1γ).

WT and FV242AA pUL21-$H_6$ were purified at pH 8.5 and PP1γ(7–300)-$H_6$ was purified at pH 7.5. Lysis buffer (20mM Tris, 20mM imidazole, 500mM NaCl, 0.5 mM $MgCl_2$, 1.4mM β-mercaptoethanol, 0.05% TWEEN-20) was supplemented with 200–400 U bovine DNase I (Sigma) and 200 μL EDTA-free protease inhibitor cocktail (Sigma), the buffer for PP1γ being additionally supplemented with 1 mM $MnCl_2$. Cleared lysates were incubated with Ni-NTA Agarose (Qiagen), washed with wash buffer (20 mM Tris, 20mM imidazole, 500 mM NaCl, plus 1 mM $MnCl_2$ [PP1γ only]) and eluted using elution buffer (20 mM Tris, 250 mM imidazole, 500 mM NaCl, plus 1 mM $MnCl_2$ [PP1γ only]). SEC was performed using a HiLoad Superdex S200 [pUL21] or S75 [PP1γ] 16/600 columns (GE Healthcare) equilibrated in 20 mM Tris, 500 mM NaCl, 1 mM DTT.

## Recombinant protein purification following mammalian cell expression

To transfect Freestyle 293F cells, 100 mL of suspension cultures at $1\times10^6$ cells/mL were transfected by mixing 100 μg of CERT pcDNA-SW with 150 μg of 25 kDa branched polyethylenimine (PEI; Sigma) in PBS for 20 min at room temperature and applying dropwise to the cells. After culturing for 72 h, cells were harvested by centrifugation (220 g, 5 min, 5˚C) and washed three times with ice-cold PBS before being resuspended in ice-cold lysis buffer with phosphatase inhibitors (100 mM Tris pH 8.0, 150 mM NaCl, 0.5 mM EDTA, 1 mM DTT, 10 mM tetrasodium pyrophosphate, 100 mM NaF, 17.5 mM β-glycerophosphate). Cells were lysed by passage through a 23G needle six times, lysates were clarified by centrifugation (40,000 g, 30 min, 4˚C) and the supernatant was passed through a 0.45 μm syringe filter (Sartorius). StrepII-CERT was captured using 1 mL StrepTrap HP column (GE Healthcare) that had been preequilibrated in wash buffer (100 mM Tris pH 8.0, 150 mM NaCl, 0.5 mM EDTA, 1 mM DTT). After extensive washing (20 column volumes) the protein was eluted using wash buffer supplemented with 2.5 mM desthiobiotin. Pooled eluate was applied to a Superose 6 10/300 SEC column (GE Healthcare) column equilibrated in 20 mM Tris pH 7.5, 150 mM NaCl, 1 mM DTT and fractions containing StrepII-CERT as assessed by SDS-PAGE were pooled, concentrated, snap-frozen in liquid nitrogen and stored at -70˚C.

## GST pull-down

Bait protein (pUL21-GST) was diluted to 1 μM in pull-down buffer (20 mM Tris pH 8.5, 200 mM NaCl, 0.1% NP-40, 1 mM DTT, and 1 mM EDTA) and, for each experiment, 200 μL of bait mixture was incubated for 30 min at 4˚C with 5 μL of glutathione magnetic beads (Pierce) that had been pre-equilibrated in pull-down buffer. Beads were washed three times with pull-down buffer and then incubated at room temperature for 60 min with 5 μM of purified prey protein in pull-down buffer, final volume of 200 μL. Beads were washed three times with pull-down buffer, bound protein was eluted using pull-down buffer supplemented with 50 mM reduced glutathione, and samples were analysed by SDS-PAGE using InstantBlue Coomassie stain (Expedeon) for visualisation.

## Mutagenesis of viral genomes and generation of recombinant HSV-1

All HSV-1 strain KOS viruses were reconstituted from a bacterial artificial chromosome (BAC) [88] and mutated viruses were generated using the two-step Red recombination method [37] and the primers shown in S2 Table. The pUL21 deletion mutant (ΔpUL21) was generated by inserting three tandem stop codons in frame after pUL21 residue 22. The pUS3 deletion mutant (ΔpUS3) was generated by removing base pairs 211–1111, which results in a stop codon after residue 70. To generate the P0 stock, Vero cells were transfected with the recombinant BAC DNA together with pGS403 encoding Cre recombinase (to excise the BAC cassette) using TransIt-LT1 (Mirus) according to the manufacturer's instructions. After 3 days the cells were scraped into the media, sonicated at 50% power for 30 s in a cup-horn sonicator (Branson), and titrated on Vero cell monolayers. The subsequent stocks (P1 to P3 and H2) were generated by infecting either Vero (P1 to P3) or HaCaT pUL21 cells (H2) at MOI of 0.01 for 3 days, after which, the cells were scraped and isolated by centrifugation at 1,000 g for 5 min. Pellets were resuspended in 1 mL of complete DMEM supplemented with 100 U/mL penicillin, 100 μg/mL streptomycin and freeze/thawed thrice at -70˚C before being aliquoted, titered on Vero cell monolayers, and stored at -70˚C until required. The presence of the desired mutations in the reconstituted virus genomes was confirmed by sequencing the pUL21 or pUS3 genes as applicable.

## Virus infections

For infections, confluent monolayers of the indicated cells were infected by overlaying with the indicated viruses, diluted in complete DMEM supplemented with 100 U/mL penicillin, 100 μg/mL streptomycin to the specified MOI. The time of addition was designated 0 hpi. After adsorption for 1 h at 37˚C in a humidified 5% $CO_2$ atmosphere, where the tissue culture plate housing the monolayers was rocked every 15 min, complete DMEM supplemented with 100 U/mL penicillin, 100 μg/mL streptomycin was added to dilute the inoculum five-fold. Infected cells were incubated at 37˚C in a humidified 5% $CO_2$ atmosphere and harvested at the specified times. For infection experiments performed in Figs 1 and 3 and 4 and S3, virus stocks propagated in Vero cells for up to four passages were used unless otherwise noted. For all other infection experiments, viruses had been propagated in Vero cells for three (P3) or four (P4) passages, or in HaCaT cells stably expressing pUL21 for two passages (H2), as indicated.

## SDS-PAGE of phosphorylated proteins from cultured cells

For analysis of protein phosphorylation, infected cells and monolayers of Vero, HaCaT or HaCaT-pUL21 cells were prepared as described above. Cells were washed two times with ice-cold 50 mM Tris pH 8.5, 150 mM NaCl and scraped into 100 μL of ice-cold lysis buffer (50mM

Tris pH 8.5, 150mM NaCl, 1% (v/v) Triton-X100, 1% (v/v) EDTA-free protease inhibitor cocktail (Sigma), 10 mM tetrasodium pyrophosphate, 100 mM NaF and 17.5 mM β-glycerophosphate). Cells destined for lambda phosphatase treatment were resuspended in lysis buffer supplemented with 1 mM $MnCl_2$ and lacking phosphatase inhibitors tetrasodium pyrophosphate, NaF and β-glycerophosphate. After 30 min incubation on ice, the lysates were sonicated (two 15s pulses at 50% power) in a cup-horn sonicator (Branson) before being cleared by centrifugation at 20,000 g for 10 min. To *in vitro* dephosphorylate control samples, cleared lysates were incubated with 400 U of lambda phosphatase (New England Biolabs, P0753S) for 1 h at 30˚C. For all samples, protein concentrations were determined using a BCA assay (Peirce) and equal amounts of protein lysates were analysed by SDS-PAGE. For enhanced separation of phosphorylated proteins, the gels contained 100 μM $MnCl_2$ and 50 μM PhosTag reagent (Wako) where indicated. After electrophoresis, PhosTag gels were soaked in transfer buffer (25 mM Tris, 192 mM glycine, 20% (v/v) ethanol) supplemented with 1 mM EDTA for 10 min with gentle rocking followed by 10 min incubation in transfer buffer alone. Separated proteins were then transferred to nitrocellulose membranes using the Mini-PROTEAN system (Bio-Rad) and analysed by immunoblotting. For quantitation of phosphoform relative abundance, immunoblots were imaged using an Odessy CLx Imaging System (LI-COR) and the signal in each lane was measured along the direction of migration as a rectangular section using the 'Plot Profile' tool in Fiji [89,90] with constant path length and rectangle width for all samples.

## Multi-angle light scattering (MALS)

MALS experiments were performed at room temperature by inline measurement of static light scattering (DAWN 8+, Wyatt Technology), differential refractive index (Optilab T-rEX, Wyatt Technology), and 280 nm absorbance (Agilent 1260 UV, Agilent Technologies) following SEC at a flow rate of 0.5 mL/min. The sample (100 μL of 1 mg/mL UL21-$H_6$) was injected onto a Superdex 200 10/300 column (GE Healthcare) pre-equilibrated in 20 mM Tris pH 7.5, 200 mM NaCl and 1 mM DTT. Molar masses were calculated using ASTRA6 (Wyatt Technology) assuming a protein *dn*/*dc* of 0.186.

## Small angle X-ray scattering

SAXS experiments were performed in batch mode at EMBL-P12 bioSAXS beam line (PETRAIII, DESY, Hamburg, Germany) [91,92]. Scattering data ($I(s)$ versus $s$, where $s = 4\pi\sin\theta/\lambda$ nm$^{-1}$, $2\theta$ is the scattering angle, and $\lambda$ is the X-ray wavelength, 0.124 nm) were collected from a sample of pUL21-$H_6$ and a corresponding solvent blank (20 mM Tris pH 8.5, 500 mM NaCl, 3% (v/v) glycerol, 1 mM DTT). The scattering profiles of the protein were measured at 1.11 and 1.66 mg/mL (30 μL sample at 20˚C; 1 mm pathlength), in continuous-flow mode, using an automated samples changer. The sample and buffer measurements were measured as 100 ms data frames on a Pilatus 6M area detector for total exposure times of 2.1 and 2.6 s, respectively. The processing and analysis of the SAXS output, including all Correlation Map (CorMap) calculations [93], were performed using the *ATSAS* 3.0.2 software package [94]. The SAXS data reported here were obtained by averaging the two data sets in PRIMUS [95]. The extrapolated forward scattering intensity at zero angle, $I(0)$, and the radius of gyration, $R_g$, were determined from the Guinier approximation ($\ln I(s)$ vs $s^2$, for $sR_g < 1.15$). The maximum particle dimension, $D_{max}$, was estimated based on the probable distribution of real-space distances $p(r)$ which was calculated using GNOM [96]. A concentration-independent estimate of molecular weight was calculated using a Bayesian consensus method [97]. All structural parameters are reported in S1 Table. The SAXS data measured for each individual concentration, with an

accompanying report are made available in the Small Angle Scattering Biological Data Bank (SASBDB) [98] entry SASDKW8.

The structural model of the ensemble of pUL21 was generated by the program EOM [36] using the experimental data processed as described above, over a range of $0.08 < s < 2.16$ nm$^{-1}$. A pool of 10,000 random conformers corresponding to pUL21N (PDB ID 4u4h) and pUL21C (PDB ID 5ed7) domains [24,25] joined by a disordered 80 amino acids linker and with a C-terminal sequence GQSVGSKH$_6$, corresponding to the C-terminal pUL21 residues not observed in the structure plus the purification tag, was generated by the software. A genetic algorithm was then employed to select ensembles of conformers from the pool with $R_g$ and $D_{max}$ distributions that best represent the experimental SAXS profile (lowest $\chi^2$). Images of the representative protein models from the refined EOM pool were generated using an open-source build of PyMOL version 1.9 (Schrödinger).

## Sequence alignment and conservation analysis

Protein sequences of pUL21 homologues from representative α-herpesviruses were as follows (UniProt ID): HSV-1 pUL21 (F8RG07), HSV-2 pUL21 (G9I242), Cercopithecine alphaherpesvirus 2 (CHV-2) pUL21 (Q5Y0T2), Saimiriine herpesvirus 1 (SaHV-1) pUL21 (E2IUE9), Bovine herpesvirus (BHV-1) pUL21 (Q65563), Equine herpesvirus 1 (EHV-1) pUL21 (P28972), Suid alphaherpesvirus 1 (PRV) pUL21 (Q04532), Anatid alphaherpesvirus 1 (AHV-1) pUL21 (A4GRJ2), Varicella-zoster virus (VZV) pORF38 (Q6QCT9), Meleagrid herpesvirus 1 (MeHV-1) pUL21 (Q9DPR5). The sequences were aligned with ClustalW [99] and conservation scores [100] were calculated using Jalview [101].

## Differential scanning fluorimetry

Differential scanning fluorimetry experiments were performed using Viia7 real-time PCR system (Applied Biosystems) and 1× Protein Thermal Shift dye (Applied Biosystems). In all experiments, assay buffer (20 mM Tris pH 8.5, 500 mM NaCl, 1 mM DTT) was mixed with dye stock solution and protein solution in an 8:1:1 ratio to give 1 ng protein in a final volume of 20 μL and experiments were performed in triplicate. Samples were heated from 25 to 95˚C at 1 degree per 20 s and fluorescence was monitored at each increment. The melting temperature ($T_m$) is the inflection point of the sigmoidal melting curve, determined by non-linear curve fitting to the Boltzmann equation using Prism 7 (GraphPad Software).

## Plaque assays

Confluent monolayers of Vero, HaCaT or HaCaT-pUL21 cells in 6-well tissue culture plates were infected with 100 pfu of the indicated virus diluted in complete DMEM to a final volume of 500 μL. After adsorption for 1 h at 37˚C in a humidified 5% $CO_2$ atmosphere, rocking the plate every 15 min, the cells were overlaid with plaque assay media (DMEM supplemented with 0.3% high viscosity carboxymethyl cellulose, 0.3% low viscosity carboxymethyl cellulose, 2% (v/v) FCS, 2 mM L-glutamine, 100 U/mL penicillin, and 100 μg/mL streptomycin) and incubated for a further 2 days. Cells were fixed with 3.7% (v/v) formal saline for 20 min, washed three times with PBS and incubated for 1 h with mouse anti-gD (LP2) [86], diluted 1:50 in blocking buffer (PBS supplemented with 1% (w/v) BSA and 0.1% TWEEN-20). After three PBS washes, cells were incubated with HRP-conjugated rabbit anti-mouse antibody (DaKo, P0161) diluted 1:5000 in blocking buffer for 30 min, followed by two PBS washes and one wash with ultrapure water. Plaques were visualized using the TrueBlue peroxidase substrate following the manufacturer's instructions (Seracare). Plaques were scanned at 1200 dpi and plaque areas were measured using Fiji [89,90].

## Single-step (high MOI) virus growth assays

To analyse virus growth curves, monolayers of the indicated cells (Vero, HaCaT or HaCaT-pUL21) were infected with the indicated viruses diluted in complete DMEM to an MOI of 3. The time of virus addition was designated 0 hpi. After adsorption for 1 h at 37°C in a humidified 5% $CO_2$ atmosphere, rocking the plates every 15 min, extracellular viral particles were neutralized with an acid wash (40 mM citric acid, 135 mM NaCl, 10 mM KCl, pH 3.0) for 1 min and then washed thrice with PBS before being overlaid with complete DMEM. At appropriate times post-infection, cells were harvested by freezing the plate at -70°C. When all plates were frozen, samples were freeze/thawed one more time before scraping and transferring to 1.5 mL tubes and stored at -70°C until titration. Titrations were performed on monolayers of Vero cells. Serial dilutions of the samples were used to inoculate the cells for 1 h, followed by overlaying with DMEM containing 0.3% high viscosity carboxymethyl cellulose, 0.3% low viscosity carboxymethyl cellulose, 2% (v/v) FBS, 2 mM L-glutamine, 100 U/mL penicillin, and 100 μg/mL streptomycin. After 3 days, cells were fixed in 3.75% (v/v) formal saline for 20 min, washed with water and stained with 0.1% toluidine blue.

## In vitro dephosphorylation assays

Dephosphorylation assays were performed upon 0.5 μM of purified $CERT^P$ or $eIF2\alpha^P$ [102] using indicated concentrations of GST-PP1 in the presence or absence of 2 μM pUL21-$H_6$ or pUL21(FV242AA)-$H_6$. Reactions (50 μL) proceeded in assay buffer (150 mM NaCl, 20 mM Tris pH 8.5, 1 mM $MnCl_2$) for 30 min at 30°C before being stopped by the addition of 50 μL of 2× SDS-PAGE loading buffer and boiling at 95°C for 5 min. Samples were analysed by SDS-PAGE using 7% (w/v) acrylamide (CERT) or 15% (w/v) acrylamide (eIF2α) gels supplemented with 25 μM PhosTag reagent and 50 μM $MnCl_2$ and the protein was visualized using InstantBlue Coomassie stain (Expedeon). To measure the ratio of $CERT^O$ to total CERT, Coomassie-stained gels were scanned using an Odyssey CLx Imaging System (LI-COR). The signal detected in the 700 nm channel for the $CERT^O$ band alone and for all CERT bands ($CERT^O$ + $CERT^P$) was quantitated using Image Studio Lite software (LI-COR) with local background subtraction.

## Virus sequencing

For Sanger sequencing of pUL21 PP1-binding mutants, the pUL21 gene was amplified from virus stock by PCR with the oligonucleotide primers 5′-ATGGAGCTTAGCTACGCCAC-3′ and 5′-TTTATTGGGGTCTTTTACACAGACTGTC-3′ using KOD polymerase (Merck) according to the manufacturer's instructions. Sanger sequencing of the amplified PCR products was performed using the sequencing primers 5′-CGTTTCCTCGCACTTTG-3′ and 5′-ACCCACGTACGGCACGTACCTC-3′.

HSV-1 genomic DNA was prepared for next-generation sequencing from partially purified nucleocapsids [103,104]. 2× T150 flasks of ~70% confluent Vero cells ($1.2 \times 10^8$ cells) were infected at an MOI of 0.1 as detailed above and incubated at 37°C in a humidified 5% $CO_2$ atmosphere until ~80% of cells showed cytopathic effect (4–6 days). Cells were harvested by scraping into medium, pelleted by centrifugation and lysed by resuspending in 5 mL of TNE buffer (50 mM Tris pH 7.5, 100 mM NaCl, 10 mM EDTA). Lipid envelopes were removed by adding an equal volume of Vertrel XF (1,1,1,2,3,4,5,5,5-decafluoropentane, Sigma-Aldrich) and mixing by inversion. The aqueous and hydrophobic phases were separated by centrifugation at 1500 g for 5 min and the aqueous phase (supernatant) was re-extracted with an additional 5 mL of Vertrel XF, mixed by inversion and separated by centrifugation. Approximately 2.8 mL of the aqueous (supernatant) phase was layered over a two-step glycerol gradient (1.3

mL of 45% (v/v) and 1.3 mL of 5% (v/v) glycerol in TNE buffer) in 5 mL thin-walled Ultraclear centrifuge tubes (Beckman Coulter). Viral nucleocapsids were pelleted by centrifugation at 23,000 rpm in an SW 55 Ti rotor for 1 h at 4˚C. The glycerol was removed and the pellet was drained by inverting the tube before being resuspended in 400 μL sterile PBS. DNA was extracted from the enriched viral nucleocapsids using the Isolate II genomic DNA kit (Bioline) according to the manufacturer's instructions, eluting in 100 μL water. DNA integrity was confirmed by agarose gel electrophoresis and next-generation sequencing of the purified viral DNA was performed by MicrobesNG (Birmingham, UK).

Reads were trimmed by MicrobesNG using Trimmomatic [105] and quality control was performed using FastQC v0.11.9 [106]. Trimmed reads from the wild-type sample were mapped using BWA mem v0.7.17-r1188 with the default settings [107] to the sequence of HSV-1 strain KOS (GenBank accession ID JQ673480.1) [38]. The sequence was then manually updated to include all variants present at >50% frequency in the wild-type sample, thus presumably being present in our cloned HSV-1 BAC. Reads from all samples were then mapped to this updated reference sequence, using the settings above and excluding all but one copy of each genomic repeat region. Reads were filtered using samtools v1.9 [108] to leave only reads which were properly paired and uniquely mapped with a mapping quality $\geq$ 10. Read coverage of the reference genome was calculated using samtools mpileup with the default settings plus '-aa -C 50 -q 10 -Q 10' to output all positions, adjust mapping quality using mismatches and exclude reads with a minimum mapping quality < 10 and positions with a base quality < 10. Variants were identified using bcftools v1.9 [108] mpileup with the same settings. Variants were filtered and quantified using the Python packages pyvcf (v0.6.8) [109] and pysam (v0.15.4) [110]. Variants were filtered to keep only positions with $\geq$ 10 reads total coverage, $\geq$ 5 reads non-reference coverage of the variant and $\geq$ 5% non-reference proportion of reads at the variant position. Variants classified as "high frequency" were those present in $\geq$ 20% of reads. Mutation types (synonymous, non-synonymous, frameshift or in-frame indel) were determined based on the gene model in GenBank accession JQ673480.1, updated to match the wild-type sequence as described above, and nucleotide positions mentioned in the text correspond to the numbering of the JQ673480.1 reference genome. Sequencing data have been deposited in the ArrayExpress database (http://www.ebi.ac.uk/arrayexpress) under the accession numbers E-MTAB-10788.

A structural model of pUS3 was generated using trRosetta [39] via the Robetta web server. Atomic coordinates for the model are available from the University of Cambridge Apollo repository (https://doi.org/10.17863/CAM.66938).

## Immunocytochemistry

Cells grown on #1.5 coverslips were infected at an MOI of 1 as listed above. At 10 hpi, cells were washed with PBS and incubated with cold 4% (v/v) electron microscopy-grade formaldehyde (PFA, Polysciences) in PBS for 15 min. Coverslips were washed with blocking buffer (1% (w/v) BSA in PBS) and permeabilized by incubation with 0.1% Triton-X for 10 min, followed by washing with blocking buffer. Primary antibodies (above) were diluted in blocking buffer and incubated with coverslips for 1 h. Coverslips were washed three times with blocking buffer before incubation for 45 min with the relevant secondary antibodies (above) diluted in blocking buffer. Coverslips were washed ten times in PBS and ten times in ultrapure water before mounting on slides using Mowiol 4–88 (Merck) containing 200 nM 4′,6-diamidino-2-phenylindole (DAPI). Images were acquired using a Zeiss LSM780 confocal laser scanning microscopy system mounted on an AxioObserver.Z1 inverted microscope using a 64× Plan Apochromat objective (NA 1.4).

## Supporting information

**S1 Fig. Potential RVxF motifs of pUL21.** (**A**) N-terminal and (**B**) C-terminal domains of
pUL21 are shown in cartoon representation with a semi-transparent molecular surface. Residues that match the consensus RVxF motif are shown as sticks with orange carbon atoms. For
both domains, key hydrophobic residues in the potential RVxF motif are buried in the hydrophobic core of the protein and would be unable to interact with the RVxF-binding pocket of
PP1 without very significant structural rearrangement.
(TIF)

**S2 Fig. Purification of hyperphosphorylated CERT.** (**A**) SEC elution profile of Strep-II
tagged CERT purified from Freestyle 293F cells grown in suspension culture. Following affinity purification using StrepTrap resin the protein was injected onto a Superose 6 10/300 column (GE Healthcare), from which it elutes as a single peak. (**B**) Purified CERT sample was
dephosphorylated by incubation with lambda phosphatase (λ phos) in the presence of $Mn^{2+}$
before being subjected to PhosTag SDS-PAGE analysis (25 μM PhosTag) and Coomassie staining, the PhosTag reagent acting to retard the electrophoretic mobility of phosphoproteins. The
lambda phosphatase treated CERT (CERT$^O$) migrates as a single band with significantly higher
electrophoretic mobility than the untreated sample, confirming that CERT purified from suspension mammalian cells is hyperphosphorylated (CERT$^P$).
(TIF)

**S3 Fig. Adaptation of viruses encoding pUL21 PP1-binding mutants to culture in Vero
cells.** (**A**) Monolayers of HaCaT or HaCaT pUL21 cells were infected with 100 pfu of indicated
viruses. Infected cells were overlaid with medium containing 0.6% carboxymethyl cellulose
and incubated for 48 h before fixing and immunostaining with chromogenic detection. Relative plaque areas (pixels) were measured using Fiji. Mean plaque sizes (red bars) were compared to WT using one-way ANOVA with Dunnett's multiple comparisons test (n = 59–119;
ns, non-significant; *, $P > 0.05$; **, $P < 0.01$; ***, $P < 0.001$). (**B**) Plaque images used for quantitation in (**A**) and Fig 4A.
(TIF)

**S4 Fig. Whole genome sequencing of wild-type and mutant HSV-1.** (**A**) Number of mapped
reads across the HSV-1 genome is shown, with repeat regions excluded from the mapping
highlighted in grey. (**B**) Regions of the HSV-1 protein coding regions where sufficient read
depth ($\geq 10$ high quality reads) was obtained to analyse sequence variants. Black ticks denote
nucleotide positions and alternating background colouring corresponds to HSV-1 genes,
green denoting overlapping reading frames. (**C**) Percentage coverage of the HSV-1 genome
(black) and coding regions (aqua) with $\geq 10$ high quality reads.
(TIF)

**S5 Fig. pUL21 antagonises pUS3-mediated protein phosphorylation and NEC distribution
in HSV-1 infected HaCaT cells.** (**A**) HaCaT cells were infected and analysed as in Fig 6A. For
the Akt sub immunoblot two exposures are shown, separated by a line, and phosphorylated
Akt substrates more abundant in cells infected with virus lacking pUL21 that can recruit PP1
(ΔpUL21 H2 and pUL21$^{FV242AA}$H2) are marked with arrowheads. (**B**). HaCaT cells were
infected at MOI = 1 with WT or mutant HSV-1, prepared as in Fig 6A. Cells were fixed at 10
hpi and stained using an antibody that recognise pUL34 (green) plus DAPI (blue). The scale
bar represents 10 μm. (**C**) HaCaT cells were infected at MOI = 5 with WT or mutant HSV-1 as
listed. Lysates were harvested at 16 hpi and subjected to SDS-PAGE plus immunoblotting
using the antibodies listed. The upper strips of the pUL31 and pUL34 blots depict SDS-PAGE

where PhosTag reagent was added to enhance separation of hyperphosphorylated (pUL31$^P$ and pUL34$^P$) and hypophosphorylated (pUL31$^O$ and pUL34$^O$) forms of the proteins. Non-specific bands are indicated with an asterisk ($^*$).
(TIF)

**S1 Table. SAXS data collection and analysis parameters.** Program version numbers are shown in parentheses.
(DOCX)

**S2 Table. Primers used to make mutant HSV-1 strains by two-step Red recombination.** Regions of the primer homologous to pUL21 are shown in *italic*, with homologous regions that will recombine being underlined. Mutations are in **bold** and the region homologous to pEPkan-S is in lower case.
(DOCX)

**S1 Data. Identification of putative pUL21 cellular interaction partners by quantitative mass spectrometry.** SILAC ratios and statistical analysis of proteins quantified by affinity capture of GFP-tagged pUL21 and quantitated by mass spectrometry (IP-MS).
(XLSX)

## Acknowledgments

We thank Stacey Efstathiou for help with monoclonal antibody generation, Kate Heesom (University of Bristol) for performing mass spectrometry analysis, Richard Roller (University of Iowa Health Care), Bruce Banfield (Queen's University), Chris Boutell (MRC-University of Glasgow Centre for Virus Research) and John Willis (University of Pennsylvania) for kindly providing antibodies, Rachel Ulferts (Francis Crick Institute) for the psPAX.2 and pMD2.G plasmids, and David Ron (University of Cambridge) for providing the GST-PP1 plasmid, phosphorylated eIF2α and helpful discussions. We thank the Cambridge NIHR BRC Cell Phenotyping Hub for assistance with cell sorting.

## Author Contributions

**Conceptualization:** Colin M. Crump, Stephen C. Graham.

**Data curation:** Tomasz H. Benedyk, Julia Muenzner, Katherine Brown, Andrew E. Firth, Cy M. Jeffries, Stephen C. Graham.

**Funding acquisition:** Andrew E. Firth, Colin M. Crump, Stephen C. Graham.

**Investigation:** Tomasz H. Benedyk, Julia Muenzner, Viv Connor, Yue Han, Katherine Brown, Kaveesha J. Wijesinghe, Yunhui Zhuang, Susanna Colaco, Guido A. Stoll, Owen S. Tutt, Stanislava Svobodova, Cy M. Jeffries, Stephen C. Graham.

**Project administration:** Colin M. Crump, Stephen C. Graham.

**Resources:** Dmitri I. Svergun, Neil A. Bryant, Janet E. Deane, Cy M. Jeffries, Colin M. Crump, Stephen C. Graham.

**Software:** Katherine Brown, Stephen C. Graham.

**Supervision:** Colin M. Crump, Stephen C. Graham.

**Visualization:** Tomasz H. Benedyk, Julia Muenzner, Katherine Brown, Stephen C. Graham.

**Writing – original draft:** Tomasz H. Benedyk, Stephen C. Graham.

**Writing – review & editing:** Tomasz H. Benedyk, Katherine Brown, Janet E. Deane, Cy M. Jeffries, Colin M. Crump, Stephen C. Graham.

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
