## [Decision Letter · Decision Letter 0]

7 Jul 2021

Dear Dr. Graham,

Thank you very much for submitting your manuscript "pUL21 is a viral phosphatase adaptor that promotes herpes simplex virus replication and spread" for consideration at PLOS Pathogens. As with all papers reviewed by the journal, your manuscript was reviewed by members of the editorial board and by several independent reviewers. The reviewers appreciated the attention to an important topic. Based on the reviews, we are likely to accept this manuscript for publication, providing that you modify the manuscript according to the review recommendations.

This manuscript was well revived by all the reviewers. You should simply address the minor concerns they may have had (few). There is no need to do more experiments.

Sincerely,

Neal A. DeLuca, Ph.D.

Guest Editor

PLOS Pathogens

Shou-Jiang Gao

Section Editor

PLOS Pathogens

Kasturi Haldar

Editor-in-Chief

PLOS Pathogens

orcid.org/0000-0001-5065-158X

Michael Malim

Editor-in-Chief

PLOS Pathogens

orcid.org/0000-0002-7699-2064

This manuscript was well revived by all the reviewers. You should simply address the minor concerns they may have had (few). There is no need to do more experiments.

Reviewer Comments (if any, and for reference):

Reviewer's Responses to Questions

**Part I - Summary**

Reviewer #1: This excellent paper convincingly shows that pUL21 of herpes simplex virus is an adapter that bridges PP1 to ceramide transport protein (CERT), an observation that likely activates its activity. The structure of pUL21 is loosely defined and is compatible with such an adapter, and the binding activity on both pUL21 and CERT are mapped. The activation of CERT may have implications for virus budding or envelopment activity.

Separately, the study shows that pUL21 helps decrease US3-mediated phosphorylation of the nuclear egress complex (NEC) comprising pUL31 and pUL34. Interesting data show second site mutations in US3 that arise upon passage of the UL21 virus in noncomplementing cells that restore viral replication and plaque size.

The scope of work is remarkable with structural biology, virology, and biochemistry all underlying the highly significant and strongly supported conclusions. The abstract is pleasingly understated, and the discussion concise.

Although the CERT and NEC activities might be viewed as separate studies, the fact that there is more than one substrate that can bind pUL21 to be de-phosphorylated further shows pUL21’s role as an adapter with limited specificity of substrate.

I do not have any significant concerns that need to be addressed. This is an important piece of work that will likely have implications for, and develop an appreciation for, an underappreciated regulatory mechanism.

Reviewer #2: In this work the authors report on the function of the UL21 protein of herpes simplex virus. Previous reports have already described its interaction with UL16, and separate possible roles via an interaction with UL11 and gE, among other possible functions. Here the authors describe and interaction, initially via transient transfection of the protein into 293 cells, between UL21 and the catalytic subunit of the cellular phosphatase, PP1. In the same analysis they also describe an interaction with the protein CERT, involved in ceramide-transport.

They then go on to identify specific regions and determinants within pUL21 for interaction required for PP1 direct binding. Following that they construct viral mutants that either lack pUL21 entirely or contain disruptions of the PP1 binding site and characterise plaque formation and replication parameters. Finally they demonstrate that while pUL21 mutants that lack the PP1 binding site initially grow extremely poorly, such mutants rapidly revert to a virtually normal phenotype. But intriguingly from the sequence analysis they show that the reversion is not due to reversion with pUL21 itself but rather due to mutation in the pUS3 kinase gene and disruption of its kinase activity. This provides very convincing evidence for a functional interaction between pUL21 and its PP1 binding activity on the one hand, and pUS3 kinase activity on the other. With convincing analysis of one of the main pUS3 targets, that is pUL34 and its role in nuclear egress, the authors provide very convincing evidence for a role of pUL21 in modulating and indeed tuning the phosphorylation of pUL34 to optimise nuclear egress activity.

Altogether this is an extremely convincing and interesting piece of work. The data are very sound, quantitatively rigorous and overall provides strong support both for the relevance of pUL20/PP1 interaction generally and more specifically for a role of this complex in moderating the pUS3 kinase activity on certain of its substrates, particulalry pUL34.

Minor points are as follows.

1. The authors report the direct binding of pUL21 and PP1 initially in transient transfection assays in 293 cells and also by interaction with purified proteins expressed in bacteria. But nowhere did I see evidence for the direct interaction by co-precipitation in virus-infected cells, and this would be useful.

2. Associated with the point above would be the simple description of the total abundance of PP1, and likely all of its isoforms, during viral infection, for example in a time course by Western blotting. The same could be said of CERT.

3. The initial finding was for a direct interaction with the catalytic subunit of PP1. It would be worth discussing whether the regulatory subunits were also co-precipitated, whether they think pUL21 interacts the holo- enzyme, or just the catalytic subunit, and since the catalytic subunit is present in very numerous combinations of holoenzymes, how would an interaction with the catalytic subunit be substrate specific.

4. For the enhancement of phosphatase activity on CERT, in figure 3E, where the authors state the range of PP1 concentrations, they don't date the stepwise increments which would be useful to try and assess quantitatively the fold enhancement by adding pUL21. It is noteworthy that this is in the absence of regulatory subunits and one wonders what the effect would be then, and how the 2uM pUL21 compares with eg addition of regulatory subunits

Overall this is a very good piece of work not only for the detail on mechanistic insight into pUL21, but also for the principal of one viral protein, via recruitment of the cellular phosphatase, fine tuning the activity of another viral protein in this case kinase. This point is made all the more convincing by in vivo analysis of viral mutants and the convincing demonstration of the revertants in a different gene.

Reviewer #3: Benedyk et al., present interesting and compelling data showing that the HSV UL21 protein interacts with and acts as an adapter for protein phosphatase 1 (PP1). They identify a motif in pUL21 that mediates interaction with PP1 and show that mutation of this motif impairs virus growth and spread, and furthermore that minimal selection of virus that have mutations in this motif yields viruses that both replicate and spread better than the parental mutant. Selection is correlated with appearance of mutations in the US3 coding sequence, suggesting that loss of pUS3 kinase activity may compensate for loss of phosphatase activity on specific substrates, including proteins of the nuclear egress complex. This is a very interesting paper, and well worth publication in this journal. I have some criticisms with regard to specific pieces of data and interpretation as listed below.

1. The authors analysis of variants that arise in response to pUL21 mutation is based on sequencing of the variant populations rather than individual, plaque-purified viruses. This has the advantage that they can sample a lot of the variation in the population without sequencing many viruses. The disadvantage is that they end up with no idea whether any of these variants is sufficient to suppress the UL21 mutant phenotype. For example, it is possible that the US3 mutations are each paired with something else that is necessary for the suppressive effect. The possibility of interactions between different mutations within these populations might have been addressed directly by generation of a recombinant double UL21/US3 mutant. While these data are sufficiently interesting without this, the authors should discuss the possibility of multiple mutation interactions and appropriately soften their conclusions about US3 involvement in genetic compensation.

2. There are two instances where the authors make semi-quantitative conclusions from western blots where quantitation of multiple experiments is not presented. The first of these is in lines 203-204, referring to Figure 3E. The authors state that “Addition of purified wild-type pUL21-H6 dramatically lowers the concentration of GST-PP1γ required for efficient CERTP dephosphorylation but addition of pUL21FV242AA-H6 does not …”. The differences pointed to are not dramatic and are, in fact, subtle enough that quantitation is required. The second instance is in Figure 6C, where the differences between pUL31 phosphorylation in WT and UL21 mutant conditions is also subtle enough to require quantitation to support the statement that the extent of hyperphosphorylation of pUL31 is increased in these conditions.

3. In lines 326-27 the authors state that The subcellular localization of pUL21 was broadly similar between this mutant and the wild-type virus, confirming that the conserved motif is not required for recruitment of pUL21 to the nuclear rim…” Recruitment of pUL21 to the nuclear rim is not at all evident in the lower two rows of panel B. In fact, magenta staining is essentially invisible in these images. The authors need better images or rigorous quantitation of the nuclear envelope localization by co-localization with pUL34 or with lamins.

Minor point:

In Figure 4E, the authors indicate the positions of residues S207 and D208 in pUS3. I think they mean to label them D207 and S208.

**Part II – Major Issues: Key Experiments Required for Acceptance**

Reviewer #1: none

Reviewer #2: This is an excellent piece of work. I see no major issues

Reviewer #3: Quantitation of western blot data in Figures 3E and 6C.

**Part III – Minor Issues: Editorial and Data Presentation Modifications**

Reviewer #1: none

Reviewer #2: 1. Providing evidence for the direct interaction by co-precipitation in virus-infected cells, and this would be useful.

2. Associated with this would be the simple description of the total abundance of PP1, and likely all of its isoforms, during viral infection, for example in a time course by Western blotting. The same could be said of CERT.

3. The initial finding was for a direct interaction with the catalytic subunit of PP1. It would be worth discussing whether the regulatory subunits were also co-precipitated, whether they think pUL21 interacts the holo- enzyme, or just the catalytic subunit, and since the catalytic subunit is present in very numerous combinations of holoenzymes, how would an interaction with the catalytic subunit be substrate specific.

4. For the enhancement of phosphatase activity on CERT, in figure 3E, where the authors state the range of PP1 concentrations, they don't date the stepwise increments which would be useful to try and assess quantitatively the fold enhancement by adding pUL21. It is noteworthy that this is in the absence of regulatory subunits and one wonders what the effect would be then, and how the 2uM pUL21 compares with eg addition of regulatory subunits

Reviewer #3: (No Response)

PLOS authors have the option to publish the peer review history of their article (what does this mean?). If published, this will include your full peer review and any attached files.

Reviewer #1: **Yes: **Joel Baines

Reviewer #2: No

Reviewer #3: No

Figure Files:

Data Requirements:

Reproducibility:

References:

---

## [Editor Report · Decision Letter 1]

23 Jul 2021

Dear Dr. Graham,

We are pleased to inform you that your manuscript 'pUL21 is a viral phosphatase adaptor that promotes herpes simplex virus replication and spread' has been provisionally accepted for publication in PLOS Pathogens.

Best regards,

Neal A. DeLuca, Ph.D.

Guest Editor

PLOS Pathogens

Shou-Jiang Gao

Section Editor

PLOS Pathogens

Kasturi Haldar

Editor-in-Chief

PLOS Pathogens

orcid.org/0000-0001-5065-158X

Michael Malim

Editor-in-Chief

PLOS Pathogens

orcid.org/0000-0002-7699-2064
---

## [Editor Report · Acceptance letter]

11 Aug 2021

Dear Dr. Graham,

We are delighted to inform you that your manuscript, "pUL21 is a viral phosphatase adaptor that promotes herpes simplex virus replication and spread," has been formally accepted for publication in PLOS Pathogens.

Best regards,

Kasturi Haldar

Editor-in-Chief

PLOS Pathogens

orcid.org/0000-0001-5065-158X

Michael Malim

Editor-in-Chief

PLOS Pathogens

orcid.org/0000-0002-7699-2064